# The Effect of Applied Potential on the Li-mediated Nitrogen Reduction Reaction Performance

Boaz Izelaar [1], Pranav Karanth[2], Arash Toghraei [3], Santosh K. Pal[2], Nandalal Girichandran[1], Mark Weijers[2], Ruud W. A. Hendrikx[4], Fokko M. Mulder [2] & Ruud Kortlever [1] ✉

The Li-mediated nitrogen reduction reaction (Li-NRR) has been proposed as one of the most promising ambient production routes for green ammonia. However, the effect of the applied potential ($E_{we}$) on the reaction performance and the properties of the solid electrolyte interphase (SEI) remain poorly understood. Herein, we combine potential controlled experiments using a reliable $Li_xFePO_4$ based reference electrode with post-mortem SEI characterization techniques, wherein we observe both an increase in the LiF concentration in the SEI, originating from LiTFSI decomposition, and the Faradaic efficiency ($FE_{NH3}$) with an increasing $E_{we}$. The transition from a predominantly organic SEI at low $E_{we}$ (−3.2 $V_{SHE}$) to a LiF-enriched layer at higher $E_{we}$ indicates the existence of kinetic barriers for the SEI formation reactions. Moreover, thicker and denser SEI structures observed at a higher $E_{we}$ enhance the Li-NRR by improving the mass transport regulation between reactant species. However, these thicker and denser SEI morphologies lead to current instabilities due to dynamic SEI thickening and breakdown.

Ammonia production based on the electrochemical synthesis of ammonia has been proposed as a promising sustainable route towards the decarbonization of the ammonia sector[1,2]. The most successful electrochemical ammonia synthesis pathway at ambient conditions is based on the non-aqueous Li-mediated nitrogen reduction reaction (Li-NRR), which was initially studied in the 1990s by Tsuneto et al.[3,4], but regained new scientific interest. Measurements performed with isotope labelled $^{15}N_2$ gas by independent laboratories have irrevocably confirmed that ammonia originates from the Li-NRR and not from external sources[5,6], which has been a major issue for the NRR in aqueous electrolytes[7–10]. Although the full mechanism is still unresolved, there is common consensus that electroplated $Li^0$ spontaneously dissociates $N_2$ into $Li_3N$, and undergoes several hydrolysis steps with a proton source (such as EtOH) via a $Li_xN_yH_z$ complex to form $NH_3$ (see Fig. 1)[11–13].

In analogy to Li-metal batteries, electroplated $Li^0$ reacts instantaneously with elements in the surrounding electrolyte, forming a layer of insoluble and partially soluble reduction products. This electronically insulating layer of solidified electrolyte shields $Li^0$ from the surrounding electrolyte, but is at the same time ionically conductive for $Li^+$ [14]. The composition of the solid electrolyte interphase (SEI) depends on the specific reduction reaction that is being favoured on the electrode surface, which correlates with the overpotential, their charge transfer kinetics and the $Li^+$ ion solvation environment[14–16].

Theoretical work suggests that the Li-NRR elementary reaction steps are fast due to the very negative potentials applied for Li plating

[1]Process and Energy Department, Faculty of Mechanical Engineering, Delft University of Technology, Leeghwaterstraat 39, 2628 CB Delft, The Netherlands. [2]Chemical Engineering Department, Faculty of Applied Sciences, Delft University of Technology, 2629 HZ Delft, The Netherlands. [3]Énergie, Matériaux, Télécommunications Research Centre, Institute National de la Recherche Scientifique (INRS), 1650 Bd. Lionel-Boulet, Varennes, Quebec J3X 1P7, Canada. [4]Material Science and Engineering Department, Faculty of Mechanical Engineering, Delft University of Technology, 2628 CB Delft, The Netherlands. ✉e-mail: R.Kortlever@tudelft.nl

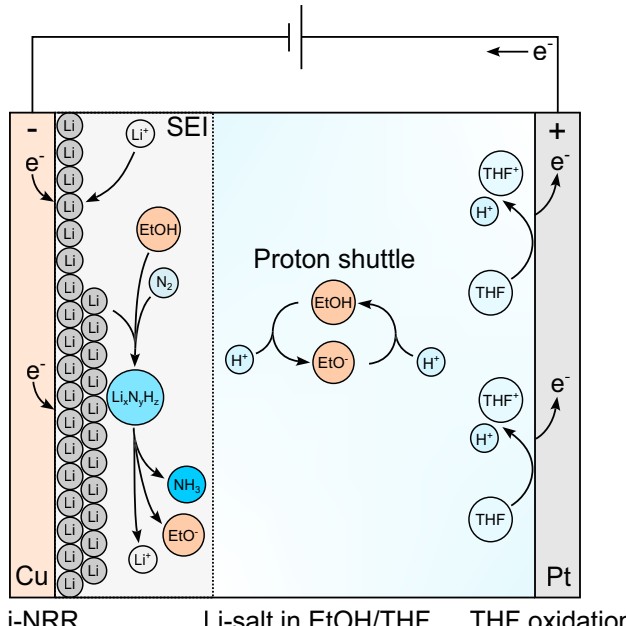

**Fig. 1 | Schematic of the Li-mediated nitrogen reduction reaction in a batch-type cell.** Although most of the reaction steps have been identified, the protonation mechanism of $Li_3N$ remains disputed, but ammonia formation seems to occur via a $Li_xN_yH_z$ complex within the SEI[12,56]. The protons for the reaction are directly supplied by EtOH and indirectly by THF oxidation at the anode via the proton shuttle[29,44].

below −3 V versus the standard hydrogen electrode (SHE), meaning that the diffusion of reactant species ($Li^+$, $N_2$ and $H^+$) through the bulk and the SEI are the rate limiting step[12,17]. When studying the reaction at atmospheric $N_2$ pressure, the ammonia Faradaic efficiencies ($FE_{NH3}$) and production rates ($R_{NH3}$) are limited by the $N_2$ transport due to its low solubility[18]. This is typically circumvented by operating at higher pressures in an autoclave cell or by implementing a gas diffusion electrode to minimize the diffusion boundary layer thickness[4,12,19,20]. The $N_2$ flux must however be balanced by the $H^+$ flux (related to the EtOH concentration) to prevent excessive lithium nitridation ($H^+$ limited regime) or EtOH hydrolysis ($N_2$ limited regime). Additionally, the diffusion rates of both $N_2$ and $H^+$ should not be significantly lower than $Li^+$ to prevent unselective lithium deposition. Hence, to reach the optimal conditions for the Li-NRR, one must carefully balance the transport rates of the reactant species by optimizing the reactant concentration and the properties of the SEI.

Chorkendorff, Norskov and coworkers estimated the $Li^+$ diffusivity rates through different SEI components by using first principle simulations. Their results suggest that the $Li^+$ transport rate through LiF is significantly lower in comparison with $Li_2CO_3$[17], while the rates of $N_2$ and $H^+$ are less dependent on the composition. Experimental results point out that LiF enriched SEI's derived from 2 M $LiBF_4$ and 0.17 M EtOH in THF or 2 M Li bis(trifluoromethanesulfonyl)imide (LiTFSI) in 0.1 M EtOH/THF can obtain a $FE_{NH3}$ above 95% under 15−20 bar $N_2$ pressure[6,17]. This is in great contrast with $LiClO_4$/LiCl enriched SEIs, where significantly lower $FE_{NH3}$'s are obtained at similar reaction conditions. It is hypothesized that LiF-enriched SEI's slow down the $Li^+$ electrodeposition rate to give the $N_2$ more time to adsorb and dissociate on the $Li^0$ active sides before another electron is consumed by Li plating.

Despite all the recent progress in Li-NRR performance, the current understanding of the reaction mechanism and specifically its potential dependency remains limited. The latter is inherently related to the commonly implemented quasi reference electrodes (QREs) such as a Ag or Pt wire, to either measure or control the potential during an experiment. These QREs have an ill-defined redox potential and are unstable under harsh non-aqueous environments, causing the potential to "drift" enormously during an electrochemical experiment[21,22]. Recently, Tort et al.[22] and McShane et al.[21] identified independently a (partially) delithiated sheet of $Li_xFePO_4$ (LFP) as a reliable reference electrode (RE) material for Li-NRR systems as its redox potential is stable over a large range of lithiation states, and it is proven to be resistant to dynamic non-aqueous environments. Both reports present a detailed proof of concept for the LFP-RE, but did not extend it further to study any existing correlations between the potential and the Li-NRR performance.

Herein, we for the first time implement a reliable LFP based RE in a high performance autoclave cell with a fluorine-based electrolyte to investigate the relationship between the applied potential ($E_{we}$) and the Li-NRR performance indicators, such as the $R_{NH3}$, $FE_{NH3}$ and reaction stability. This allows us to identify the individual voltage contributions of a batch-type Li-NRR system and use these to optimize the energy efficiency (EE). To probe any potential induced effects on the SEI composition and morphology, we perform post-measurement characterization techniques including X-ray photoelectron spectroscopy (XPS), scanning electron microscopy (SEM) and solid-state nuclear magnetic resonance spectroscopy (ssNMR), wherein the latter has never been implemented in the context of Li-NRR. Electrolyte concentration effects are also included in this study to analyse any existing correlations between the electrolyte concentrations and the $E_{we}$.

We identify three potential regimes, in which the current response up to −3.2 V (all reported potentials are in V *vs* SHE) is the most stable, but at the cost of a relatively low $FE_{NH3}$ (< 22%) and $R_{NH3}$ (< 16 nmol $s^{-1}$ $cm^{-2}$). At more negative potentials (down to −4.0 V), both the $FE_{NH3}$ and $R_{NH3}$ increases to ~50% and 350 nmol $s^{-1}$ $cm^{-2}$, respectively. Beyond −4.0 V, breakdown of the current response is initiated, while the $FE_{NH3}$ reaches to a maximum of 62.9 ± 2.2% at −4.6 V. SEI characterization results show higher LiF concentrations at more negative $E_{we}$, indicating that a significant overpotential is required to overcome the fluorine-based electrode decomposition barrier. The strongest positive correlation was observed between the $FE_{NH3}$ and the LiF concentration. Thicker and denser SEI morphologies observed at −3.7 V and −4.6 V are also beneficial for the $FE_{NH3}$, while they can be responsible for the observed current instabilities beyond −4.0 V. These findings improve the current understanding of the SEI formation process and sheds light on a new optimization strategy for Li-NRR systems, which contribute to the development of a sustainable ammonia production process.

## Results

### Electrochemical characterization of the Li-NRR system

All Li-NRR experiments were carried out in a batch-type three-electrode autoclave cell (Fig. 2a and Supplementary Fig. 1), with a Cu wire as a working electrode, Pt wire as counter electrode and a delithiated sheet of $Li_xFePO_4$ as reference electrode (LFP-RE). The main part of the analysis will be done with 2 M LiTFSI dissolved in 0.1 M EtOH/THF as this electrolyte was earlier identified by Du et al. as a high-performance electrolyte[6]. Additionally, we performed Li-NRR experiments with 1 M LiTFSI to analyse any possible correlations between the electrolyte concentration and the $E_{we}$. The operating pressure of the autoclave cell was kept constant at 20 bar $N_2$ pressure to enhance the $N_2$ transport rate and is in-line with other studies that used an autoclave cell (Supplementary Table 1). The LFP-RE was prepared by delithiating a sheet of $LiFePO_4$ (LFP) via Li stripping against a coiled Cu wire in a two-electrode configuration (Fig. 2b) until ~50 mol% of LFP was converted to $FePO_4$ (FP). X-ray diffraction with Rietveld refinement of a freshly prepared LFP-RE (Fig. 2c) indicates two clear crystalline phases of 67 mol% LFP and 33 mol% FP, respectively. Although this deviates to some extent from an ideal 50/50 molar distribution, the LFP/FP molar

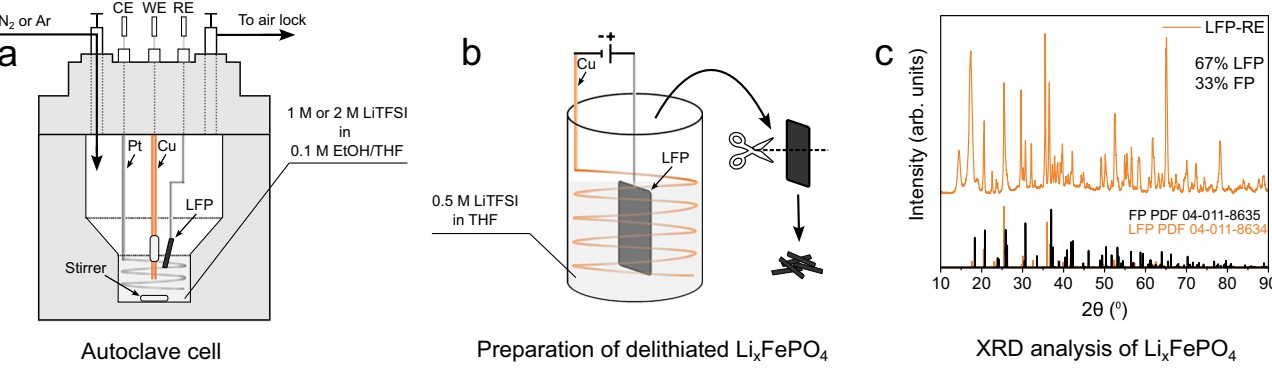

**Fig. 2 | Preparation and verification of the Li$_x$FePO$_4$ reference electrode. a** Schematic of the autoclave cell. **b** Illustration of the delithiation procedure of a LFP sheet. **c** XRD analysis of the LFP-RE with reference entries of LFP and FP. Fitting of the Rietveld refinement can be found in Supplementary Fig. 2.

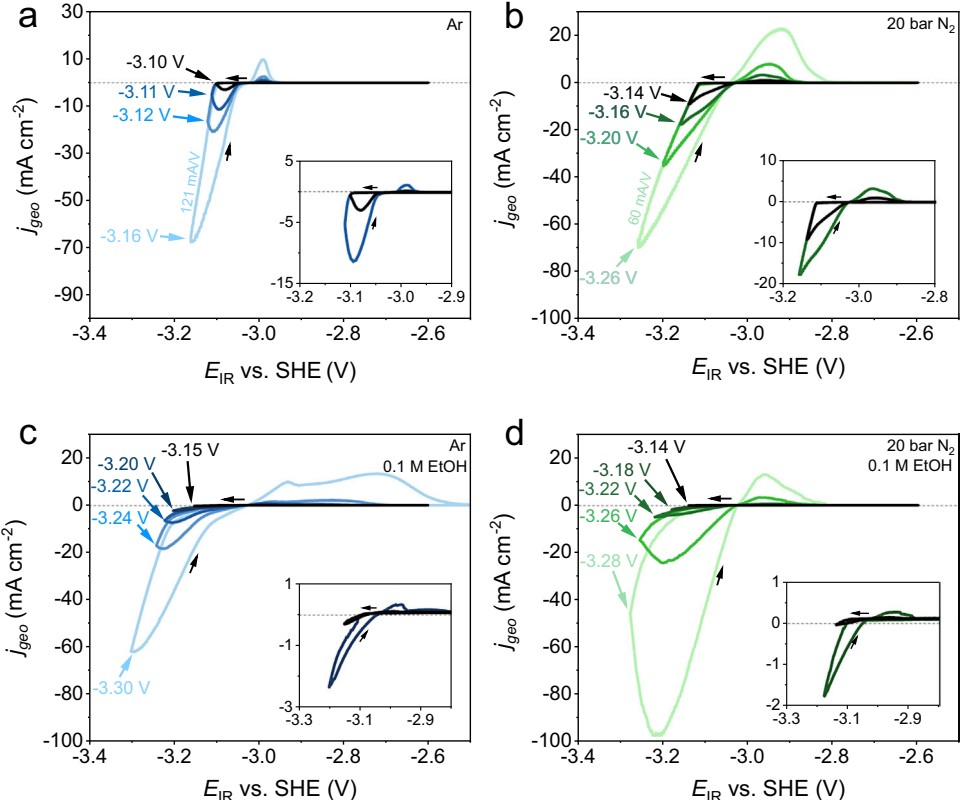

**Fig. 3 | Cyclic voltammetry under different operation conditions.** Stable cyclic voltammograms with different reduction potentials (indicated by the arrows) are plotted from black to light blue (Ar) or green (N$_2$). **a** 2 M LiTFSI in THF under Ar. The inset shows the CVs with reduction potentials until -3.10 V and −3.11 V. **b** 2 M LiTFSI in THF under 20 bar N$_2$. The inset shows the CVs with reduction potentials until −3.14 V and −3.16 V. **c** 2 M LiTFSI in 0.1 M EtOH in THF under Ar. The inset shows the CVs with reduction potentials until −3.15 V and −3.20 V. **d** 2 M LiTFSI in 0.1 M EtOH in THF under 20 bar N$_2$. The inset shows the CVs with reduction potentials until −3.14 V and −3.18 V. All measurements were conducted at room temperature, a scan rate of 20 mV s$^{-1}$ and 85% $IR_u$ drop corrected.

ratio of our LFP-RE is sufficient, since the Fe$^{2+}$/Fe$^{3+}$ redox potential of ~0.4 V *vs* SHE is stable within a broad range of lithiation states[21].

The LFP-RE was used to establish current density - voltage (*j-E*) relationships of the Li-NRR system with 2 M LiTFSI under Ar (Fig. 3a) and 20 bar N$_2$ pressure (Fig. 3b), with and without EtOH (Fig. 3c, d). The main aim of these measurements is to investigate whether the overpotential of Li$^+$ reduction is influenced by different species in the electrolyte, and to identify or exclude the presence of other (electro) chemical reactions. To that end, we performed multiple cyclic voltammetry (CV) experiments around -3 V and monitored the reduction and oxidation peaks by gradually shifting the reduction potentials to more negative values. At the start of each measurement, the Cu wire

was preconditioned by scanning for at least 20 cycles at 20 mV s$^{-1}$ between −3.10 V and -2.60 V. Afterwards, the *j-E* behaviour was stable and reproducible. All voltammograms indicate a Li/Li$^+$ equilibrium potential ($E_{Li/Li^+}$) between -3.03 V and -3.02 V (Fig. 3), which is within close proximity of the standard equilibrium potential in THF (-2.98 V)[15,23]. This small discrepancy is not related to a malfunctioning LFP-RE, but is assigned to differences in the activity coefficient of Li$^+$ ions in the solution due to a different salt or solvent selection and salt concentration[23]. A minimum overpotential of 0.08 V was necessary to initiate Li nucleation on the Cu wire, irrespective of the addition of EtOH or N$_2$. However, the overall *j-E* relationship is significantly influenced by species other than the Li-salt. The integrated charge of the

reduction ($Q_{red}$) and oxidation ($Q_{ox}$) peaks of the CVs (Supplementary Fig. 3) reveal a striking degree of asymmetry. Voltammograms with 2 M LiTFSI in an Ar atmosphere (Fig. 3a) show a clear Li$^+$ reduction peak, while Li$^0$ oxidation is mostly absent. We assign this behaviour to the SEI formation process, whereby the majority of "freshly" plated Li$^0$ reacts instantaneously and irreversibly via non-Faradaic reactions with nearby solvent molecules (TFSI$^-$ and THF). Therefore, an initial part of the electrons is always lost due to SEI formation until the entire layer of metallic Li is electronically insulated from the surrounding solvent molecules. Thus, the degree of reversibility between the Li plating and stripping process can in principle be used to evaluate whether the SEI structure is fully developed (steady state). Even after a significant number of cycles, it was not possible to obtain a reversible Li plating/stripping peak under these conditions, which could be related to the relatively fast scanning rate (20 mV s$^{-1}$) that disrupts the formation of a stable SEI[24].

After pressurizing the cell with N$_2$, the Li plating overpotential ($\eta_{Li}$) increases by roughly 0.1 V (at -70 mA cm$^{-2}$) in Fig. 3b. The $\eta_{Li}$ can be approximated by the Doyle-Fuller-Newman model[23,25]:

$$\eta_{Li} = \varphi_s - \varphi_e - \rho_{SEI} L_{SEI} \frac{j_{int}}{a_-} \tag{1}$$

Where $\varphi_s$ and $\varphi_e$ are the solution and electrode potential, $L_{SEI}$ the SEI thickness, $\rho_{SEI}$ the resistivity of the SEI, $a_-$ the specific interfacial area of the cathode and $j_{int}$ the interfacial current density. The third term in Equation (1) is related to SEI characteristics, which suggests that the presence of nitrogen in the SEI (most likely in the form of Li$_3$N) has a significant effect on the SEI's properties[26]. We expect that the heterogeneity of multiple SEI species may result in a geometric expansion of $L_{SEI}$, which increases the $\eta_{Li}$. Similarly, previous literature reports observed an increase in the $L_{SEI}$ when EtOH is present in the electrolyte[27,28]. This falls in line with our voltammograms (Fig. 3a, c), where the $\eta_{Li}$ increases by -0.15 V (at -60 mA cm$^{-2}$) when using the EtOH containing electrolyte under Ar atmosphere. Interestingly, the voltammograms of the Li-NRR system (containing N$_2$ and EtOH, Fig. 3d) show a distinct $j$-$E$ relationship. When cycling between -3.26 V and -2.60 V, we notice that Li plating becomes more favourable at less negative potentials and conforms with the other Li plating peaks in the absence of EtOH. The development of a secondary Li plating peak can be observed more clearly in Supplementary Fig. 4, where the electrolyte with EtOH was slowly exposed to N$_2$ gas during a CV experiment. It seems that the unique interplay of N$_2$ and EtOH during the Li-NRR at the electrode's surface does not significantly alter the Li plating overpotential.

**Relationship between the potential and the Li-NRR performance**
Chronoamperometry (CA) measurements at different $E_{we}$ (-3.0 V to -4.6 V) and salt concentrations (1 M and 2 M LiTFSI) under Li-NRR operation conditions were performed over the course of 4 h to study its impact on the reaction stability, $R_{NH3}$ and FE$_{NH3}$. Due to the high solubility of NH$_3$ in the electrolyte, the amount of volatile NH$_3$ in the headspace is typically negligible in autoclave systems[6,15]. Therefore, we did not use a downstream acid trap and decided to only consider the quantified NH$_3$ in the electrolyte to calculate the $R_{NH3}$ and FE$_{NH3}$. The stability of the LFP-RE was checked by determining the $E_{Li/Li^+}$ via cyclic voltammetry before and after a CA measurement, where we did not observe any noticeable changes (Supplementary Fig. 5). More details regarding the electrochemical measurement procedure can be found in the Method section. Based on the stability of the CA measurements (Fig. 4a), the $R_{NH3}$ and the FE$_{NH3}$ (Fig. 4b), we identified three $E_{we}$ regimes that will be further defined as a (1) low $E_{we}$, (2) moderate $E_{we}$ and (3) high $E_{we}$ regime. With 2 M LiTFSI, the current response remains very stable up to -3.2 (low $E_{we}$ regime), while the $R_{NH3}$ (< 16 nmol s$^{-1}$ cm$^{-2}$) and FE$_{NH3}$ (< 22%) are at relatively low levels.

Within the moderate $E_{we}$ regime between -3.2 V and -4.0 V, the FE$_{NH3}$ and $R_{NH3}$ gradually increases to ~50% and 350 nmol s$^{-1}$ cm$^{-2}$. Initial signs of breakdown and the complete breakdown of the Li-NRR starts to occur close to and below -4.0 V (high $E_{we}$ regime), where the $j$ starts declining after 3 min at -4.6 V. Interestingly, the FE$_{NH3}$ keeps increasing with respect to $E_{we}$. A similar relationship between the potential and the performance is observed using 1 M LiTFSI, but with significantly lower FE$_{NH3}$ and $R_{NH3}$ at all examined $E_{we}$ (see Fig. 4e, f). To verify that NH$_3$ originates from Li-NRR, an Ar blank test at -3.7 V was performed which did not result in any quantifiable NH$_3$. This demonstrates that our earlier reported cleaning procedures work effectively[8].

We observe a change of electrolyte colour with the applied potential, wherein it transitioned from being transparent (low $E_{we}$), to golden yellow and orange (moderate $E_{we}$), and to brown at high $E_{we}$ (Fig. 4c). This colour change is most likely related to the (by-)products from solvent oxidation[29], while we also suspect that small parts of the fragile SEI breaks down and dissolves into the electrolyte during the cell's operation, depressurization, transfer and handling in the glovebox. A fully dispersed SEI in the electrolyte typically gave a black and viscous appearance (Supplementary Fig. 6). We noticed, however, that for higher $E_{we}$ the SEI layer remained mostly intact when the cell was slowly depressurized and handled in a careful and controlled manner. To remove this experimental uncertainty, we modified the cell with an additional purge valve (Supplementary Fig. 1b), allowing us to remove the electrolyte before depressurization as safeguard (as was previously mentioned by Chorkendorff and coworkers)[17]. After repeating several experiments with this new approach, we obtained black appearing SEI layers (Fig. 4d). Although the shape and thickness varied to some extent with the $E_{we}$, the SEI's at low $E_{we}$ were not visible by the eye. For the Pt anode, we noticed visible adsorbates on the surface due to surface poisoning from arguably THF polymerization reactions or decomposition of the TFSI$^-$ anion[29,30]. Consequently, the anodic potential increased far beyond the $E_{eq}$ of THF oxidation (+1.00 V in non-aqueous media) and EtOH oxidation (+1.25 V in non-aqueous media) during the majority of the electrochemical measurements (Supplementary Fig. 7).

To investigate whether ethanol is lost via anodic oxidation, we analysed the electrolyte post-measurement with liquid $^1$H and $^{13}$C NMR to search for possible side products. Ethanol and THF in anhydrous environments oxidize via a carbocation reaction into H$^+$ and a carbonium cation, where the latter undergoes a nucleophilic attack with neighbouring solvent molecules[31–33]. Ethanol cations will most likely react into diethyl ether or an acetal (1,1-diethoxyethane)[31], while THF cations initiate ring opening THF polymerization reactions[32,33]. The current matter has recently been investigated by Simonov, MacFarlane and coworkers wherein they identified 2-ethoxy-tetrahydrofuran as the main side product[29]. This matches well with our NMR analysis, suggesting that ethanol oxidation does not occur but still reacts away via an outer-sphere mechanism with a THF cation to extend its ring with an ethoxy group. Our $^1$H NMR spectra (Supplementary Fig. 8) confirm this notion by showing a significantly weaker EtOH signal when a 2-ethoxy-tetrahydrofuran signal is present. Traces of ring-opening polymeric chains of THF were also observed in both $^1$H and $^{13}$C NMR spectra (Supplementary Fig. 9). The $^{19}$F NMR spectra (Supplementary Fig. 10) only shows a TFSI$^-$ signal (80 ppm), suggesting that the anion does not oxidize during an electrochemical measurement, although we cannot exclude TFSI$^-$ oxidation completely due to the detection limit. An elaborate analysis of the liquid NMR results can be found in Section S1. of the Supplementary Discussion.

**Potential effect on the composition and morphology of the SEI**
The equilibrium potential of Li plating is considered to be more negative than the reduction potential of THF (+0.50 V vs Li/Li$^+$)[34], TFSI$^-$ (+0.75 V vs Li/Li$^+$)[35], and EtOH (+1.00 V vs Li/Li$^+$)[36]. Therefore, the kinetic stability of the reactants increases in order of THF > TFSI$^-$ >

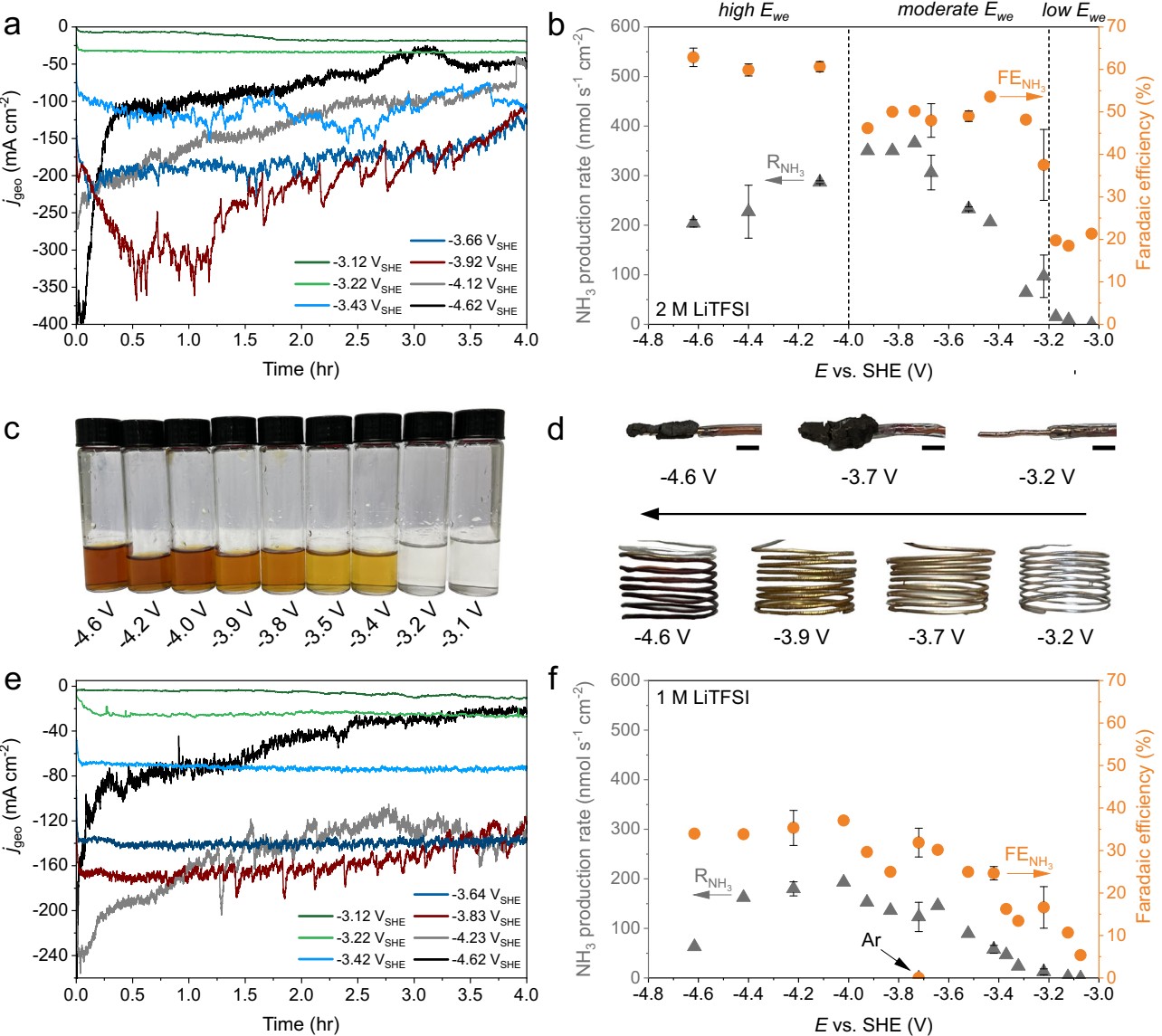

**Fig. 4 | Potential dependency of the Li-mediated nitrogen reduction performance. a** Chronoamperometry measurements using 2 M LiTFSI (with 0.1 M EtOH) as electrolyte. **b** Effect of the $E_{we}$ on the $R_{NH_3}$ and the $FE_{NH_3}$ with 2 M LiTFSI. **c** Photographs of the 2 M LiTFSI electrolyte solutions post-measurement. **d** Photographs of different Cu electrodes with and without a SEI layer and the

Pt anodes post-measurement. The black scale bar indicates a 2 mm length. **e**, **f** are analogous to (**a**, **b**) but with 1 M LiTFSI as electrolyte. All measurements were conducted at 20 bar $N_2$ pressure and room temperature. The error bars represent the mean ± standard deviation derived from two independent measurements.

EtOH, indicating that there can be a correlation between the SEI composition and the electrode potential. To further understand the earlier established relationship between the $E_{we}$ and the Li-NRR performance with 2 M LiTFSI, we carried out post-measurement characterization with SEM, XPS and ssNMR to analyse the morphology and composition of the SEI within the earlier defined potential regimes. To ensure that the SEI layer remains in-tact, we always removed the electrolyte with the purge valve before degassing the autoclave cell (Supplementary Fig. 1b). The retrieved electrodes were washed with THF to remove residual salts and subsequently dried in the glovebox.

At -3.2 V (low $E_{we}$ regime), the SEM images (Fig. 5a, b) of the Cu wire reveal a thin cracked layer, which can be identified as the SEI. These cracks are most likely the result of contraction during THF evaporation. The curvature of the wire and the cracks reveals cross-sectional views of the SEI which can be used as a qualitative indicator of the SEI thickness and reveal morphological information (see

Supplementary Fig. 11). The SEI thickness varied over the length of the Cu wire and is estimated to be between 1–7 μm. The surface under the SEI (Fig. 5c) highlights nano-spherical deposits, which extends into a chain linked macro-porous network (Supplementary Fig. 12) until the layer becomes uniformly passivated, showing a relatively smooth surface. The XPS surface scan indicates a strong carbon signal (Fig. 6a), with a prominent C-C (284.8 eV) peak in the high-resolution C 1s spectra (Fig. 6b), which is often associated with adventitious carbon. The XPS depth profile analysis reveals an elemental shift towards organic species with shorter carbon-chains and a higher oxygen content, indicating a more ethanol-derived layer. Besides Li ethoxide as being reported as the main product of EtOH, we also observe a C = O peak in the C 1s spectra, which is typically affiliated with Li carbonate and can suggest an alternative decomposition pathway. This agrees well with the O 1s spectra, where the singlet peak could originate from C-O (532.7 eV), C = O (531.5 eV) and Li alkoxide (530.4 eV) signal contributions[27,37]. Our surface structure and composition shows

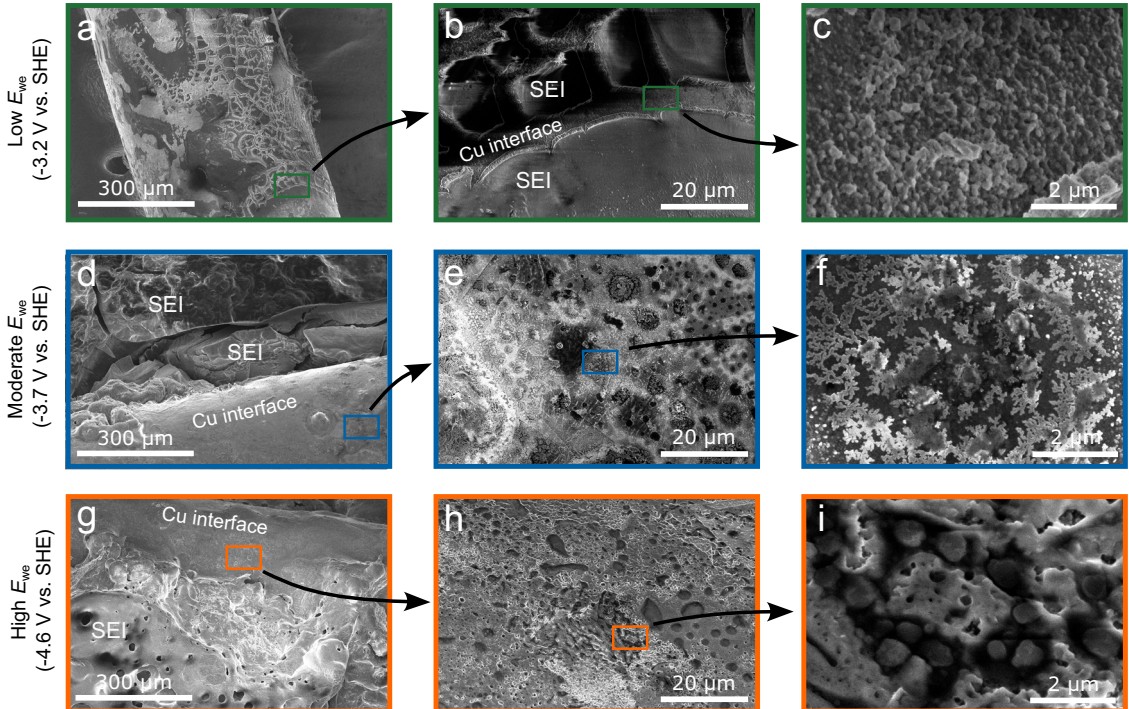

**Fig. 5 | SEM images of the Cu electrodes post-measurement.** An overview of the SEM images of the Cu electrodes showing the SEI morphology and the Cu-SEI interface. These electrodes were obtained after 4 h chronoamperometry experiments at -3.2 V with different magnifications: (**a**) 250x (**b**) 2000x (**c**) 20000x. At -3.7 V with different magnifications: (**d**) 250x, (**e**) 2000x (**f**) 20000x. At -4.6 V with different magnifications: (**g**) 250x (**h**) 2000x, (**i**) 20000x. Additional SEM images of the SEI's are available in Supplementary Figs. 11–13. 2 M LiTFSI in 0.1 M EtOH/THF was used as the electrolyte.

similarities with the work of Steinberg et al.[27], wherein they did not identify a metallic Li phase, but mostly a poor passivation layer of ethanol-derived species.

Based on the earlier mentioned equilibrium potentials, we expected that all species in the electrolyte would decompose at -3.2 V and form a mixture of both organic and inorganic SEI species, while the XPS results clearly indicate an EtOH-derived layer. We propose that the EtOH decomposition reaction on Li⁰ is preferred because it is the most kinetically unstable compound in the electrolyte. Consequently, the thin Li-alkoxy passivation layer formed upon electrode polarisation can hinder the decomposition reaction of other species in the electrolyte[34]. Although ammonia production was observed, the low $FE_{NH3}$ ( < 20%) indicates that nitrogen activation is not favourable at these conditions, which is most likely related to an imbalance in the transport rates of Li⁺, H⁺, and N₂ diffusion through the SEI layer. Since our operating conditions (N₂ pressure, EtOH and salt concentration) are relatively similar to previous literature reports observing higher $R_{NH3}$ and $FE_{NH3}$'s, we expect that the properties of the SEI at -3.2 V is mainly responsible for the low $FE_{NH3}$. Additionally, Spry et al.[38] and Benedek et al.[39] operated at ambient N₂ pressures and managed to obtain similar $FE_{NH3}$'s (~20%), and even higher (40%). Hence, increasing the N₂ operating pressure does not directly result in better Li-NRR performance. In partial agreement with the work of Chorkendorff, Norskov and coworkers[17], it seems that the SEI properties have a much stronger effect on regulating the transport rate of Li⁺ and H⁺ than on N₂. The porous Li-ethoxide structure seems to be particularly poor in slowing down the Li⁺ and H⁺ diffusion rate, leading to build-up of more SEI material (due to rapid and uncontrolled Li plating) and the formation of hydrogen gas. The latter explains the visible macro pores and cavities in the internal segment (Supplementary Fig. 12b) and top surface of the SEI (Supplementary Fig. 12c).

At more negative $E_{we}$, the SEI layers were hundreds of micrometres thick and difficult to analyse with SEM. Therefore, we decided to break the layer in a controlled manner until the Cu wire and a cross-sectional view of the deposits were exposed (Fig. 5d–i, Supplementary Figs. 13–16). This allowed us to analyse the Cu interface and the internal morphology of the SEI but did not result in a well-defined cross-sectional surface, leading to rough estimates of the SEI thickness. At -3.7 V, the Li microstructure on the Cu interface changes from particle-like to dendritic features (Fig. 5f), which can be signs of a diffusion limited growth regime[40]. The Cu interface at -4.6 V also contains Li deposits, but with a more rod-like dendritic geometry surrounded by a dense SEI layer (Fig. 5i). The elemental composition of the SEI at -3.7 V and -4.6 V (Fig. 6a) reveals an organic surface layer (most likely adventitious carbon), while the subsurface layers are predominantly inorganic with an increasing order of Li > F > C > O > S > N present. The F 1 s spectra in Fig. 6b discloses a prominent LiF peak (684.5 eV), wherein the majority of elemental fluorine is in the form of LiF via LiTFSI decomposition[41], which is in agreement with other studies employing a F-based salt[6,17]. Other LiTFSI decomposition products via its sulfone groups, such as Li₂SO₄ (166.8 eV) and a mix of Li₂Sₓ (Li₂S₆ at 162.8 eV, Li₂S₄ at 161.2 eV) and Li₂S (159.8 eV) were also identified in the S 2p spectra (given binding energies are from the S $2p_{3/2}$ orbital)[41,42], but remained in low quantities. Deconvolution of the Li 1s peak is challenging because it resembles a singlet representing all Li species with overlapping binding energies (Supplementary Fig. 17). In this work, ssNMR is used as a complementary characterization technique and is especially useful for the identification of several SEI materials in the bulk phase, such as metallic Li, LiTFSI, LiF and organic species based on their unique chemical shift. Unfortunately, the application of ssNMR was unsuccessful at -3.2 V due to the limited availability of SEI material.

At -3.7 V, the absence of a metallic Li peak in the ⁷Li NMR spectra (Fig. 7a) indicates that the layer of Li dendrites (observed by SEM), is thin and only present on the Cu interface. Based on ¹⁹F NMR (Fig. 7b), LiF (-203 ppm) represents most of the fluorine compounds in the SEI (55 ± 8%) and agrees well with our XPS results (see Supplementary Table 2). When shifting to a more negative $E_{we}$ (-4.6 V), the LiF content (74%) with respect to LiTFSI (26%) increases substantially, suggesting

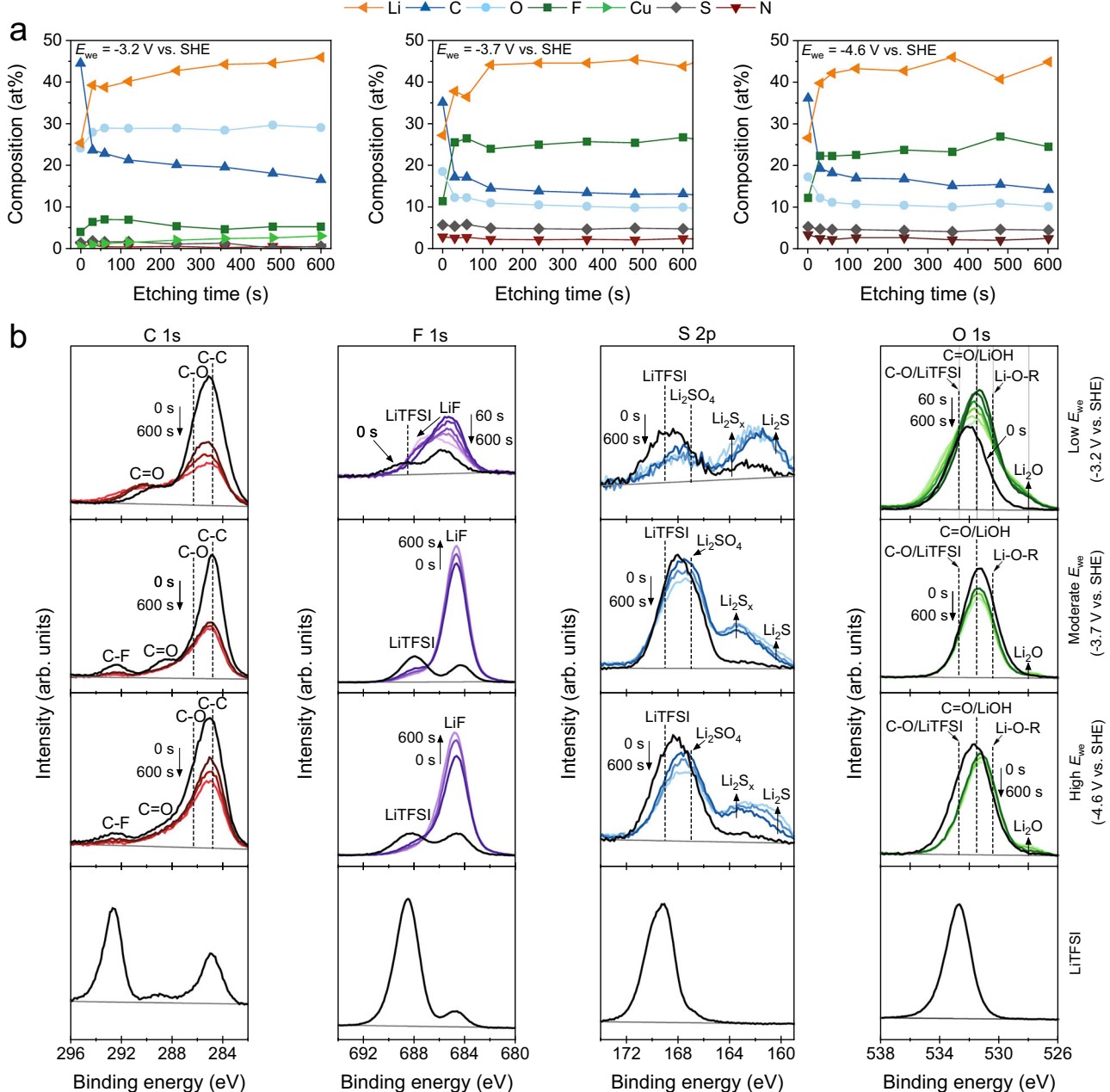

**Fig. 6 | XPS depth profiling of the solid electrolyte interphases. a** Elemental composition of the SEI obtained post-measurement at -3.2 V, -3.7 V and -4.6 V after 0-600 s of Ar⁺ etching. **b** High resolution XPS spectra of C 1s, F 1s, S 2p and O 1s of the SEI obtained post-measurement at -3.2 V, -3.7 V and -4.6 V after 0 s, 60 s, 240 s and 600 s of Ar⁺ etching. Vertical dashed lines indicate binding energies of known

chemical species. Reference lines in the S 2p spectra indicates the binding energy of the S 2p$_{3/2}$ orbital. Surface scan of LiTFSI is also included for referencing. Corresponding XPS survey spectra are available in Supplementary Figs. 18–20. 2 M LiTFSI in 0.1 M EtOH/THF was used as the electrolyte.

TFSI⁻ reduction requires a significantly higher activation barrier than the solvent decomposition reactions. Additionally, a metallic Li peak (at 265 ppm) becomes visible in the ⁷Li NMR spectra (Fig. 7a), indicating that the freshly electroplated Li⁰ does not immediately react with the electrolyte or solvent species as occurs at low $E_{we}$. Higher concentrations of LiF seems to correlate well with the existence of a metallic Li⁰ layer, and agrees with the notion that LiF has better electron insulating properties than other SEI constituents. Despite the majority of the SEI being inorganic, THF-based species are also revealed in XPS C 1s spectra (Fig. 6), ¹H NMR (Fig. 7c) and ¹³C NMR (Supplementary Fig. 21) spectra. Ethoxide-species were not detectable, suggesting that its concentration is below the limit of detection or short-lived and immediately re-dissolve back into the electrolyte to act

as a proton shuttle as was previously suggested[27,43,44]. Dissolution of Li ethoxide is further supported by the observation of a two orders of magnitude thinner SEI at -3.2 V (~1–7 μm) in comparison with the substantial inorganic layers at -3.7 V (~0.8-1 mm) and -4.6 V (~0.25–0.30 mm, Supplementary Fig. 11, 14 and 16), while the accumulated charge differs only by a factor of 6.9 and 3.3, respectively (Supplementary Fig. 7).

We find a clear correlation between the FE$_{NH3}$ and the LiF concentration in the SEI (induced by the potential driving force). LiF is not necessary to make ammonia, but simulations based on first principles have pointed out that the Li⁺ transport resistance within LiF layers is much higher in comparison to other SEI species[17]. Therefore, the Li⁺ diffusion rate (and perhaps also the H⁺ diffusion rate as was pointed

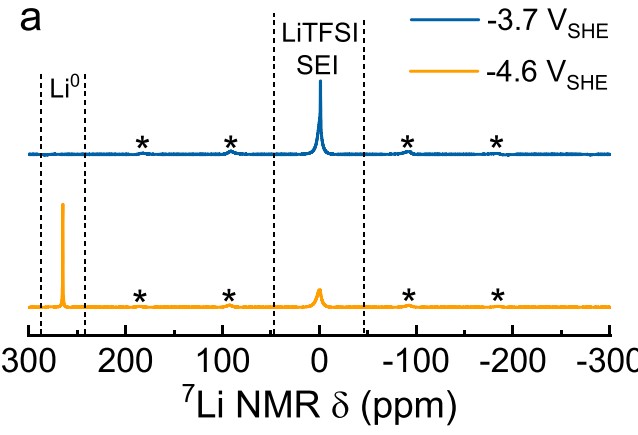

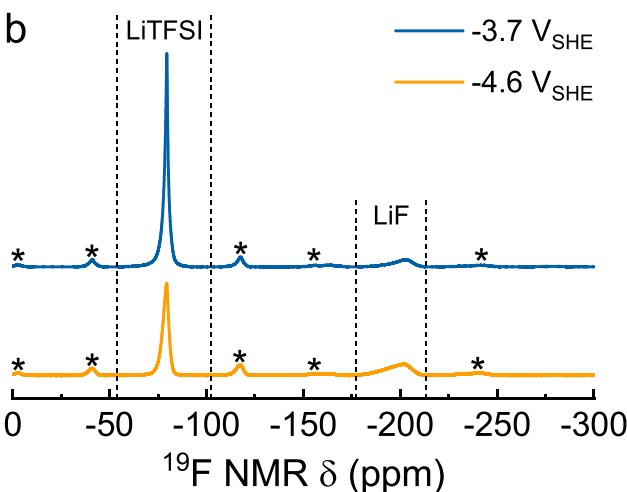

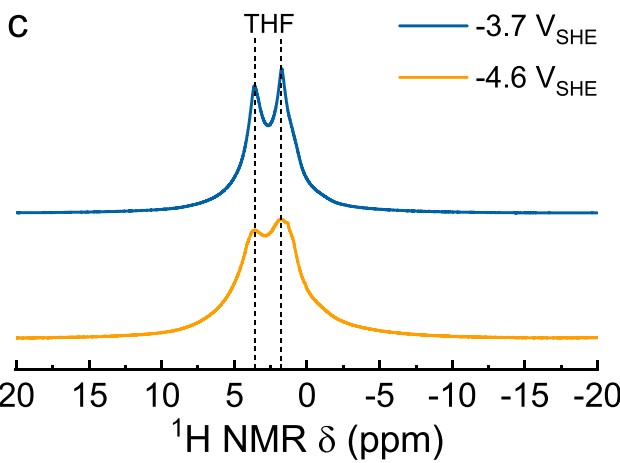

**Fig. 7 | Solid-state NMR spectra of the solid electrolyte interphases.** The SEIs for ssNMR were obtained after chronoamperometry at -3.7 V and -4.6 V. **a** $^7$Li NMR spectra with signals at 0 ppm and -265 ppm indicate LiTFSI, Li-SEI materials and metallic Li$^0$, respectively[57]. **b** $^{19}$F NMR spectra with a large peak at -80 ppm, representing a −CF$_3$ contribution from LiTFSI or derivative products. The small and broad peak at -203 ppm is attributed to LiF. **c** $^1$H NMR spectra have two broad peaks at 3.63 ppm and 1.75 ppm that match with THF. The SEI compound distribution is summarized in Supplementary Table 2. 2 M LiTFSI in 0.1 M EtOH/THF was used as the electrolyte.

out by us earlier) slows down relative to N$_2$ transport and this increases the lithium nitridation probability, leading ultimately to higher FE$_{NH3}$. These findings match well with our experimental observations, but the structure of our SEI is more complicated and most likely resembles a mixture of polyhetero organic and inorganic microphases that is typically observed in Li-ion batteries (Fig. 8)[14]. Hence, we cannot exclude the influence of other species then LiF entirely.

Thicker and especially denser SEI structures obtained at -3.7 V and -4.6 V (in comparison with -3.2 V) seem to also correlate with the FE$_{NH3}$. A similar observation was earlier reported by McShane et al.[28], wherein the FE$_{NH3}$ monotonically increases with the SEI thickness. We expect that the more tortuous paths from the bulk towards the electrode surface slows down the Li$^+$ (and probably H$^+$) transport rate, while N$_2$ is less affected. The exact mechanism at play remains unknown and requires further investigation. The current response is a direct measure of the Li plating rate and the Li$^+$ flux as Li plating is considered to be the only electrochemical reaction occurring at the working electrode. Therefore, instabilities in the current (Fig. 4a) are likely related to a dynamic process between SEI thickening and breakdown, leading eventually to an increase or decrease of the Li$^+$ transport resistance during chronoamperometry measurements.

### Electrolyte concentration effects

Both the FE$_{NH3}$ and $R_{NH3}$ dropped significantly when 1 M LiTFSI is implemented, while their respective trends with the $E_{we}$ remain relatively similar in comparison with 2 M LiTFSI. Discrepancies between the electrolyte concentration and the Li-NRR performance are most likely related to differences in the Li$^+$ solvation environment since species within the solvation shell are preferentially reduced at the electrode surface[15,16]. LiTFSI concentration effects in the context of Li-NRR performance have only been studied to a limited extent. To that end, we used Raman spectroscopy to study the coordination chemistry of the Li$^+$ - TFSI$^-$ - THF solvation environment in the bulk electrolyte. The SEI composition was also analysed with XPS, wherein we focused particularly on the high $E_{we}$ regime to compare the LiF concentrations between 1 M and 2 M LiTFSI.

The XPS results at 1 M LiTFSI (high resolution spectra in Supplementary Fig. 22 and survey scan in Supplementary Fig. 23) show a shift towards an almost evenly distributed ratio of elemental C, O and F. The C 1s spectra and O 1s spectra contain substantial C-O and C = O signal contributions, suggesting mostly ethoxy- and carbonate-based species instead of THF-decomposition products as was observed with 2 M LiTFSI in the high $E_{we}$ regime. A relative decrease in the quantity of inorganic species in the SEI could explain the lower selectivity when following a similar reasoning for 2 M LiTFSI. The overall decrease in $R_{NH3}$ is likely related to the lower availability of Li$^+$ ions for plating and FE$_{NH3}$[6].

The position of the strong S-N-S bending vibration in TFSI$^-$ in the Raman spectra is often used to study its coordination with Li$^+$ via its four available oxygen atoms. The vast amount of literature typically allocates: 738−742 cm$^{-1}$ as solvent-separated ion pairs (SSIP, uncoordinated anions), 744−747 cm$^{-1}$ as contact ion pairs (CIP, anion coordinated with a single Li$^+$), and 747−750 cm$^{-1}$ as aggregated coordination mode (AGG, anion coordinated to more than one Li$^+$)[45–47]. Figure 9a, b illustrates a clear increasing trend between the percentage of CIP interactions and the LiTFSI concentration. Moreover, the strong ring breathing vibration band (914 cm$^{-1}$) of "free" THF (Supplementary Fig. 24) reveals two prominent shoulder features (902 cm$^{-1}$ and 922 cm$^{-1}$) at elevated salt concentrations[48], and the THF ring stretching vibration at 1030 cm$^{-1}$ noticeably shifts to 1035 cm$^{-1}$. Both Raman signals indicate strong alterations in the THF symmetry due to more anion-solvent interactions (see Fig. 9c), which explains why the LiF concentration in the SEI is higher when implementing 2 M LiTFSI[49]. In contrast to LiFSI[15], AGGs were not identified within the selected

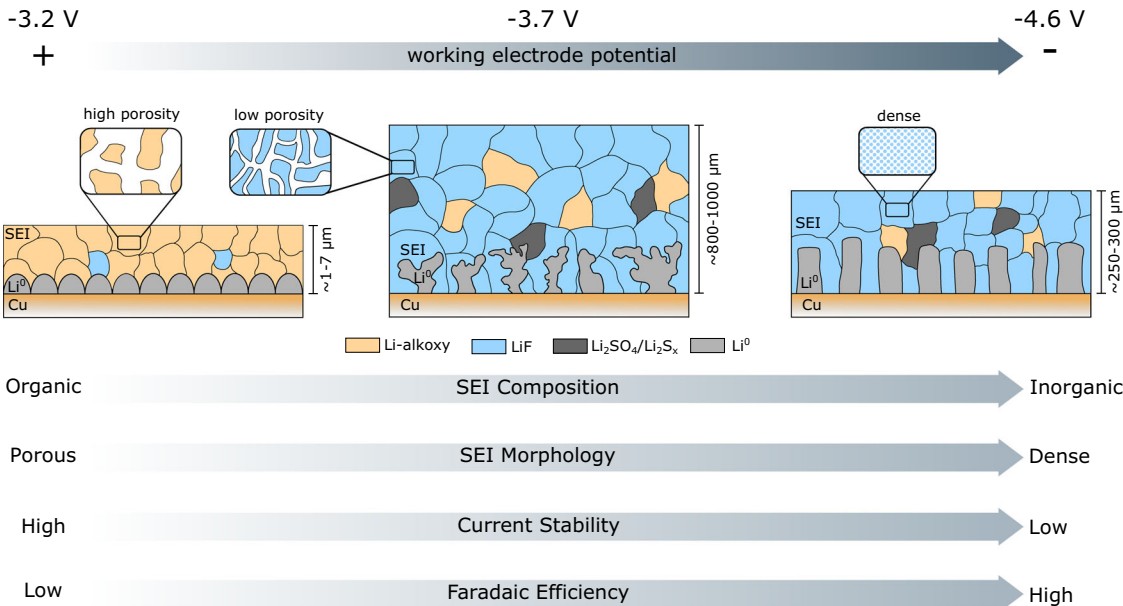

**Fig. 8 | Schematic of the SEI composition, thickness and Li morphology at different applied potentials.** At -3.2 V, the SEI was only a few μm thick, showing particle-like Li microstructure and a mostly organic composition. At -3.7, the SEI grew into a layer of hundreds of micrometer thick, consisting mostly of inorganic F-species, while the morphology of the Li deposits is mostly dendritic. At -4.6 V, the Li microstructure is also dendritic, but with a more rod-like geometry surrounded by a dense LiF-enriched SEI. The patch work of different SEI microphases resembles a mosaic pattern that is typically observed in Li-ion batteries and also in the Li-NRR context[14,27].

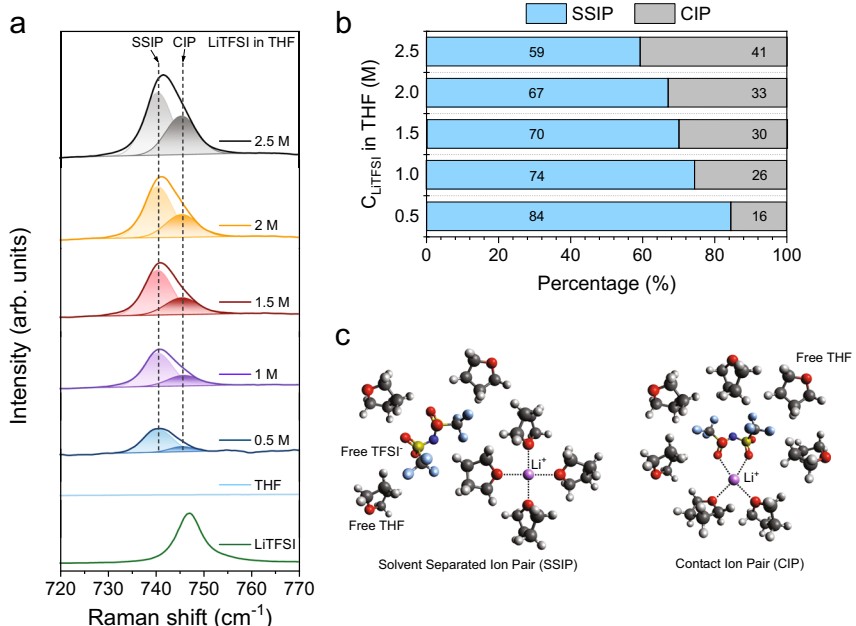

**Fig. 9 | Raman spectroscopy of different LiTFSI concentrations dissolved in THF. a** Raman shift of the S-N-S bending vibration in crystalline LiTFSI (746.8 cm⁻¹) and TFSI⁻ dissolved in THF between 0.5–2.5 M. The Raman peaks were deconvoluted into a SSIP peak (740.5 cm⁻¹) and CIP peak (745.5 cm⁻¹) in OriginPro 10. **b** Percentage between the SSIP and CIP coordination modes with the LiTFSI salt concentration. **c** Schematic of the SSIP and CIP solvation environment of the Li⁺ - THF - TFSI⁻ system based on ref. 58.

concentration range, and can be related to the salt's lower dissociation energy[15], and the strong Li⁺ ion chelating capabilities of THF via its electronegative ether functionalities[50].

## Energy efficiency of a batch Li-NRR system

The EE is a useful metric to evaluate the current state of our system in the field and to identify the most optimal potential regime. The EE was calculated with the assumptions from Chorkendorff and coworkers for their continuous Li-NRR flow cell[51]. For the batch-type systems, we slightly modified their expression by substituting the hydrogen energy input with the consumption of the sacrificial solvent (THF) as a proton source (more details can be found in Section S2. of the Supplementary Discussion). The EE of previous literature reports using a batch cell were re-estimated with this new expression (see Supplementary Fig. 25

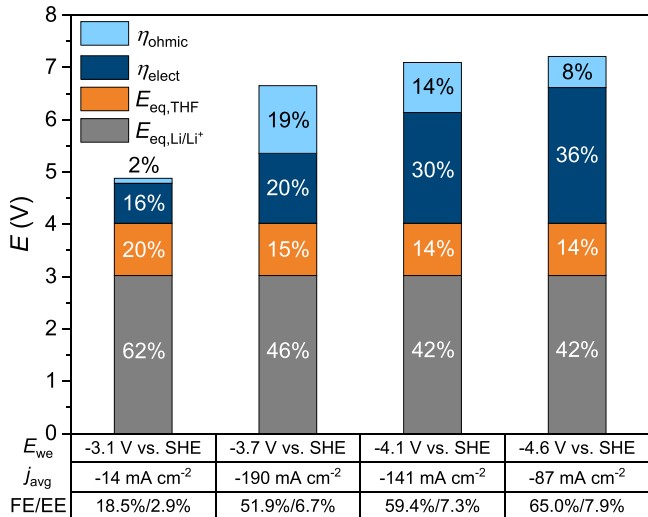

| $E_{we}$ | -3.1 V vs. SHE | -3.7 V vs. SHE | -4.1 V vs. SHE | -4.6 V vs. SHE |
|---|---|---|---|---|
| $j_{avg}$ | -14 mA cm⁻² | -190 mA cm⁻² | -141 mA cm⁻² | -87 mA cm⁻² |
| FE/EE | 18.5%/2.9% | 51.9%/6.7% | 59.4%/7.3% | 65.0%/7.9% |

**Fig. 10 | Cell voltage contributions at different applied working potentials.** Equilibrium potentials of Li⁺ reduction and THF oxidation are taken as $E_{eq,Li/Li^+}$ = −3.02 V and $E_{eq,THF}$ = +1.00 V, respectively. The ohmic overpotential is calculated via $\eta_{ohmic} = IR_\Omega$ ($R_\Omega$ = 71 Ω, Supplementary Fig. 26) and the electrode overpotential is the remaining voltage contribution after subtraction ($\eta_{elect} = E_{cell} - E_{eq} - \eta_{ohmic}$). Data is based on the chronoamperometry measurements in Supplementary Fig. 7.

and Supplementary Table 1). The maximum EE is 18% when assuming a FE of 100%, THF energy costs of 476 kJ/mol and an ideal $E_{cell}$ of -4.02 V ($E_{eq}$ of Li⁺ reduction and THF oxidation)[52].

The highest EE of 8% is obtained in the high $E_{we}$ regime (-4.6 V) and has a stronger dependency on the FE_NH3 than on the voltage losses due to the additional energy input of the sacrificial solvent. Therefore, all charge that is not allocated to nitrogen activation results in a severe energy penalty. The current response is, however, very unstable at such potentials. Hence, from a long-term operation standpoint it is more beneficial to operate in the moderate $E_{we}$ regime at the cost of only ~1% EE (Fig. 10). MacFarlane, Simonov and coworkers managed to obtain an EE of 13% when excluding the SEI formation period, as they reached a FE near unity over longer testing periods while using similar operating conditions (2 M LiTFSI in 0.1 M EtOH/THF with 15 bar N₂ pressure)[6]. We did not intend to optimize for Li-NRR performance metrics, but reach a similar FE over shorter tests where ammonia synthesis and SEI formation occur simultaneously, while remaining discrepancies may be allocated to the salt purity[11,16], hydrodynamics of the cell[6], and electrode configuration[53].

The voltage losses in our cell are related to the electrode overpotentials and ohmic losses via resistive dissipation (Fig. 10), wherein the ohmic contributions only become significant at high $j$ due to our compact cell design ($R_\Omega$ = 71 Ω, Supplementary Fig. 26) and small working electrode area (0.1 cm²). The electrode overpotentials are predominantly associated with the Li⁺ transport resistance through the SEI and THF oxidation, where the former becomes more significant in the high $E_{we}$ regimes due to the build-up of a thicker and denser SEIs. Based on the cyclic voltammograms, the Li⁺ charge transfer resistance cannot be fully excluded (~0.3 V at -45 mA cm⁻², Fig. 3d) but contributes to a lesser extent. Anodic overpotentials are related to Pt surface poisoning by organic residues from solvent oxidation, which increase the charge transfer resistance, but can be reduced by substituting the sacrificial solvent with hydrogen oxidation as a proton source. It is, however, equally important to implement a selective electrocatalyst that favours HOR in non-aqueous media (such as PtAu)[51,54]. Additionally, the side products of THF oxidation may compromise the properties of the electrolyte and the SEI, leading potentially to system instabilities. According to our liquid NMR results, EtOH is also consumed by these side products, indicating that sufficient

proton concentrations cannot be sustained for long term measurements. Hence, a Li-NRR system based on THF oxidation is most likely not technically and economically feasible, and a H source from H₂ oxidation would be preferred[55]. The low solubility of H₂ (0.78 mmol/L H₂O) is an issue for autoclave cells because the HOR will quickly become mass transfer limited. A continuous flow cell based on gas diffusion electrodes can enhance the mass transfer of N₂ and H₂ as was initially demonstrated by the Manthiram group and later upscaled by Chorkendorff and coworkers[19,51]. GDE flow cells have an inherently better EE than the batch-type cell (Supplementary Fig. 25). The latter is however still useful for fundamental studies or for a quick screening of different salts, solvents and active mediators beyond Li.

In summary, Li-NRR experiments under 20 bar N₂ pressure were for the first time performed with a reliable reference electrode based on a partially delithiated sheet of Li_xFePO₄. This allowed us to investigate the relationship between the $E_{we}$ and the Li-NRR performance indicators, such as the FE_NH3, $R_{NH3}$ and reaction stability. Additionally, SEI characterization was performed post-mortem with XPS, ssNMR and SEM to gain better insights into the underlying mechanisms. With 2 M LiTFSI and at -3.2 V, both the FE_NH3 (< 22%) and $R_{NH3}$ (<16 nmol s⁻¹ cm⁻²) remained relatively low. The SEI resembles a thin (~1–7 μm) porous layer of Li ethoxide species, which is commonly known as a poor transport regulator for the reactant species. The FE_NH3 (50%) and $R_{NH3}$ (350 nmol s⁻¹ cm⁻²) increased significantly at more negative $E_{we}$ (-3.7 V) and SEI forms a thick and dense SEI layer (~800–1000 μm) that is predominately enriched with LiF. This indicates the existence of a strong correlation between the FE_NH3 and the LiF concentration. Thicker and denser SEI morphologies are also beneficial for the FE_NH3, while we also link them to current instabilities beyond -4 V. Therefore, it is more beneficial to operate at moderate $E_{we}$ (-3.7 V) for long-term operation. At 1 M LiTFSI, the overall trend between the $E_{we}$ and Li-NRR performance is relatively similar, but there is an overall reduction in the FE_NH3 and $R_{NH3}$ over the entire potential range. This is related to the lower availability of Li⁺-TFSI⁻ contact ion pairs, leading to a lower concentration of inorganic species in the SEI. Hence, both the $E_{we}$ and the Li⁺ solvation environment play a key role in the eventual morphology and composition of the SEI. These findings improve the current understanding of the SEI formation process and showcases new optimization strategies for Li-NRR systems that contribute to the development of a sustainable ammonia production process.

## Methods
### Materials
Copper wire (Ø0.5 and Ø2 mm, 99.95%) and platinum wire (Ø0.5 mm, 99.9%) were purchased from Mateck. A sheet of double coated LiFePO₄-on-aluminium sheet (241 mm × 200 mm×0.1 mm) with a specific capacity of 127 mAh/g and coating areal density of 160 g/m² was obtained from MTI Corporation. Anhydrous tetrahydrofuran (99.9%, inhibitor) and ethanol (< 30 ppm H₂O) were supplied by Sigma-Aldrich and VWR, respectively. Li bis(trifluoromethanesulfonyl)imide (< 20 ppm H₂O, 99.9%) was purchased from Solvionic and did not require further drying. Molecular sieves (3 A 4–8 mesh, Sigma) were purchased from Sigma-Aldrich. Their activation procedure was as follows: The molecular sieves were washed with acetone, pre-dried overnight in a vacuum oven (at 80 °C), transferred to the antechamber of the Ar glovebox (GS, <0.1 ppm H₂O, <0.1 ppm O₂) and dried a second time at 200 °C for 24 h. Anhydrous EtOH and THF were dried over activated molecular sieves for 5 days with a 25% mass/volume ratio and stored over a new batch of activated molecular sieves inside the glovebox. Dimethyl sulfoxide (≥ 99.9%) and trimethylsilane (analytical grade for NMR) supplied by Sigma-Aldrich were used as received as internal standard and internal reference for liquid NMR. KBr (99%) was used as an internal filler for solid state NMR and was purchased from Sigma-Aldrich and pre-dried for at least 24 h under vacuum. Lower grade

ethanol (denatured 96%) and acetone ($\geq$99%) were used for various cleaning purposes and were supplied by Technisolv and VWR. Concentrated sulfuric acid (95–98 wt % $H_2SO_4$, trace metal purity) was bought from Sigma-Aldrich and used either directly for glassware acid cleaning or diluted for other purposes. Both potassium hydroxide (85%) and phosphoric acid ($\geq$85%) were purchased from Sigma-Aldrich. Ultrapure water (Milllipore Milli-Q 7000) was used for solution preparation and cleaning. High purity $N_2$ and Ar gases (99.999%) were supplied by Linde.

### Reference electrode preparation

A small piece (2.4 cm × 1.4 cm) was cut from a double coated LiFePO$_4$-on-aluminium sheet and mounted into a two electrode beaker cell filled with 0.5 M LiTFSI in pre-dried THF (<10 ppm) using a Cu wire (Ø2 mm) in a helix coil as anode. The electrode was partly delithiated at 0.1 C rate for 5 h (charging current is ~0.6 mA) to obtain a half charged LFP electrode with separate $LiFePO_4$ and $FePO_4$ phases, which results in a well-defined LiFe$^{2+}$PO$_4$/Fe$^{3+}$PO$_4$ redox potential of +0.4 V vs SHE[21,22]. The delithiation experiment and further storage of the delithiated sheet was done in the glovebox to prevent phase transitions during air exposure.

### Electrochemical measurements

All electrochemical ammonia synthesis experiments were performed in a polyether ether ketone (PEEK) three electrode autoclave cell at 20 bar $N_2$ pressure. The cell design and configuration was inspired by the work of MacFarlane, Simonov and coworkers[6]. The cell consists of a glass insulated Cu wire (Ø0.5 mm × 6 mm) as working electrode (WE), a Pt wire (Ø0.5 mm × 400 mm) as counter electrode (CE) coiled around the WE with a Ø14 mm, a small Li$_x$FePO$_4$ ribbon (~2 mm×11 mm x 0.1 mm) as reference electrode (RE) positioned near the WE (see Supplementary Fig. 1 for more details), and a glass magnetic stirrer (Ø5 mm × 12 mm, Fischerbrand). The insulated Cu wire was electropolished at 5 V versus the copper anode for 2 min in a two-electrode beaker cell containing $H_3PO_4$ and a Cu anode (Ø2 mm) coiled in helix shape. The smooth Cu wire (indicated by the scanning electron microscopy image Supplementary Fig. 27) was sonicated in water for 5 min and blow dried with $N_2$. The Pt wire was flame annealed and reshaped into the Ø14 mm coil. The RE was soaked in a diluted LiTFSI/THF solution for about an hour inside the glovebox for cleaning purposes. The Pt wire, stirrer and the internal body of the cell were rinsed with acetone, ethanol, acid cleaned in 10 vol% $H_2SO_4$ (95–98%, Sigma) in water for an hour, and rinsed excessively with water and blow dried with $N_2$. If the cell parts were exposed to ambient air for >1 day, an additional 15 min of sonication in 0.1 M KOH was added to the cleaning procedure to remove any surface accumulated NO$_x$ species[8]. Other items, such as the top part of the cell, o-rings (ERIKS), beakers, vials, caps and spatulas were all cleaned with ultrapure $H_2O$ and blow dried with $N_2$. The cell parts, consumables and other labware required for assembling the cell in the glovebox were dried overnight in a vacuum oven (Vacuterm, Thermo Scientific) at 90 °C and ≤3 mbar. Afterwards, all items were transferred to a preheated antechamber at 80 °C and flushed 3 × 5 min before introducing into the glovebox. A fresh batch of 1 M or 2 M LiTFSI in 0.1 M EtOH/THF was prepared prior to each experiment. Moisture content of the electrolyte was measured by Karl Fischer titration (Metrohm, 756 KF Coulometer) and was typically <5 ppm $H_2O$ (Supplementary Fig. 28). The assembled cell was transferred out of the glovebox and connected to our gas purification skid (see Supplementary Fig. 29) on the bench. Residual moisture in the gas ($\leq$ 3 ppm $H_2O$) was removed by a home-made stainless-steel column (Ø12mm x 250 mm, Swagelock) filled with activated molecular sieves. Any remaining impurities were removed (<1 parts per trillion) via a certified commercial gas filter (Entegris GPUS35FHX). The cell was slowly pressurized until 20 bar $N_2$ and saturated for at least 30 min.

The stirring rate was set to 300 rpm throughout the entire experiment.

All electrochemical measurements were performed using a SP-200 Biologic potentiostat in combination with EC-Lab software. A typical measurement sequence was as follows: (i) An initial potentiostatic electrochemical impedance spectroscopy (PEIS) measurement at open-circuit voltage (OCV) was carried out to determine the ohmic resistance ($R_u$) between the RE and the WE. (ii) Cyclic voltammetry was performed between -2.6 V and -3.2 V for 10 cycles to examine whether the Li/Li$^+$ equilibrium potential ($E_{Li/Li^+}$) is close to -3.0 V. (iii) Chronoamperometry (CA) was performed at the potential of interest for 4 h. (iv) The RE potential was reassessed using cyclic voltammetry to detect possible potential drifts. (v) A final PEIS was performed to determine any changes in the $R_u$. The applied potential was corrected post-measurement if we noticed minor deviations in the LFP-RE potential by taking the average of (ii) and (iv).

After the electrochemical ammonia synthesis experiment, the pressurized head space was slowly purged through an ethanol trap (Supelco Analytical, 6-4835) that functioned as an air lock. Subsequently, the cell was flushed for 10 min with Ar to remove left-over $N_2$ gas from the cell. The cell was disconnected and reintroduced into the glovebox to withdraw the electrolyte and remove the RE for cleaning. The rest of the cell was cleaned outside the glovebox following the procedure outlined earlier. We noticed that during depressurization, the fragile solid electrolyte interface layer breaks down and disperses into the electrolyte as was earlier observed by Chorkendorff and coworkers[17]. This rapid displacement of dissolved $N_2$ gas causes vigorous movements of the electrolyte. Therefore, we added a small drain to one of the cell bodies (Supplementary Fig. 1b) to remove the electrolyte before degassing. In order to sustain the SEI as much as possible, the electrolyte was directly removed after the CA measurement, meaning that the additional CV (iv) and PEIS (v) were not performed. After removing the electrolyte, the procedure was kept the same as before. The WE with the SEI was stored in the glovebox for further physical characterization.

### Physical characterization

Semi-quantitative information related to the phase composition of the delithiated LFP electrode was obtained by X-ray diffraction (XRD) with Rietveld refinement. XRD was performed on a Bruker D8 Advance diffractometer with a Bragg-Brentano geometry, a Lynxeye position sensitive detector, a divergence slit with a 12 mm opening (V12), a scatter screen with 5 mm height, and a Cu K$_\alpha$ ($\lambda$ = 1.5406 Å) radiation source at 45 kV 40 mA operation conditions. The measurement was done within the 5−135° 2θ range with a 1 s time per step and a 0.020° 2θ step size. Bruker DiffracSuite.EVA v6.1 was used to subtract the background, correct small displacements, strip the K$_{\alpha2}$ contribution from the patterns, and identify present phases using the ICDD pdf4 database. The Rietveld refinement was performed in Profex.

The solid-electrolyte interface (SEI) was characterized post-mortem with X-ray photoelectron spectroscopy (XPS), solid-state nuclear magnetic resonance spectroscopy (ssNMR) and scanning electron microscopy (SEM). Before characterization, the SEI was suspended in 2 mL of dried THF for a few minutes to remove any salt precipitation on the surface. For XPS, a mobile XPS sample stage with a vacuum sealable lid (Supplementary Fig. 30) was used to minimize air exposure during transfer from the glovebox to the XPS chamber. During a typical procedure, the parts of the sample holder were shortly dried in the antechamber and introduced into the glovebox. Small parts of the SEI were carefully deposited onto the sample stage. The sample holder was assembled, vacuum sealed in the antechamber and transferred into the XPS chamber. XPS spectra were acquired with a Thermo Scientific K$_\alpha$ spectrometer with a monochromatic Al K$_\alpha$ excitation source. The analysis chamber has a base pressure of about $2 \times 10^{-9}$ mbar. High resolution XPS spectra were recorded using a 400

μm spot size, 0.1 eV step size, and 50 eV pass energy (200 eV for survey). C 1s adventitious carbon (284.8 eV) was used to correct the charge of all spectra. A depth-profile of the sample was generated by $Ar^+$ ion etching (1000 eV, 2 mm × 2 mm) at different time intervals in between the XPS measurements. CasaXPS v2.3 was used to deconvolute the obtained spectra.

For ssNMR sample preparation, the entire SEI layer (7–17 mg) that was built-up during the potentiostatic measurements was scraped off from the WE with a PTFE spatula and placed inside a mortar. Between 37-47 mg of KBr was added to the mortar as an inert filler and was mixed with the SEI material using a pestle. The SEI-KBr mixture was carefully packed into a 3.2 mm diameter airtight $ZrO_2$ rotor. This rotor filling procedure was entirely executed in an Ar glovebox to prevent moisture and air exposure. All items that were used during the rotor filling procedure (including KBr) were pre-dried at 60 °C in a vacuum oven for at least 24 h before they were introduced into the glovebox for the sample preparation. All MAS NMR measurements were performed on a Bruker Ascend 500 MHz (11.7 T) spectrometer with a NEO console using a 3.2 mm triple resonance probe. $^7Li$, $^{19}F$ and $^1H$ NMR measurements were carried out at a spinning speed of 18 kHz with direct excitation pulses using a pulse length of 5 μs (45 W), 3.25 μs (107.75 W), 3.25 μs (107.75 W), a recycle delay ($D_1$) of 30 s, 10 s, 4 s, and 128, 128, 64 number of scans ($N_{scan}$), respectively. $^{13}C$ NMR measurements were decoupled from the $^1H$ signal by using the "HPDec" pulse sequence with a pulse length of 5 μs (110 W), $D_1$ of 3 s, $N_{scan}$ is 9600 scans, and a $^1H$ decoupling power of 20 W. We noticed that the $^{79}Br$ side bands of KBr interfere with the $^{13}C$ NMR measurements, therefore the spinning speed was adjusted to 20 kHz for a better peak separation. Cross-polarisation measurements between $^1H$ and $^{13}C$ NMR ($^1H$-$^{13}C$ CP) were also carried out to identify the organic contributions in the $^{13}C$ NMR spectra. The following settings were used for $^1H$-$^{13}C$ CP measurements: $^1H$ pulse length of 3.4 μs (78.5 W), $^1H$ RF field strength of 65 kHz, contact time of 500-3000 μs and a $D_1$ of 1.5 s. All $^7Li$ and $^{19}F$ NMR spectra were externally referenced to LiCl at 0 ppm and internally referenced to LiTFSI at -79.5 ppm, respectively. $^1H$ and $^{13}C$ NMR spectra were referenced to adamantane at 1.76 ppm and 37.85 ppm, respectively. All resulting spectra were processed and analysed in MestReNova 15. A line broadening of 10 Hz was applied to the $^7Li$, $^{19}F$ and $^1H$ NMR spectra, and 250 Hz to the $^{13}C$ NMR spectra.

Liquid phase NMR was also used to analyse the electrolyte solution post-measurement for any oxidation or decomposition products. All liquid NMR measurements were performed on a 400 MHz Agilent 400-MR DD2 spectrometer equipped with a 5 mm Agilent OneNMR room temperature probe. $^1H$, $^{13}C$ (with $^1H$ decoupling) and $^{19}F$ NMR measurements were executed with a $D_1$ of 2 s, 3 s, 20 s, and $N_{scan}$ of 1024, 4000 and 60 scans, respectively. Prior to use, the NMR tubes (NORELL) were always washed with water and acetone with a dedicated vacuum-based cleaning station, and dried for at least 4 h at 70 °C in an electric furnace. A mixture (50 μL) of DMSO as internal standard and TMS as internal reference were added to 450 μL of electrolyte solution and transferred into the pre-dried NMR tube. A capillary tube filled with $C_6D_6$ was added to the tube as locking solvent. All resulted spectra were processed and analysed in MestReNova 15. A line broadening of 1 Hz was applied to the $^1H$ and $^{13}C$ NMR spectra.

The morphology of the electrode-electrolyte interface was analyzed by field-emission secondary electron microscopy (FE-SEM, FEI Helios G4 CX) with through lens detector (TLD) in secondary electron (SE) mode. Imaging was performed at low accelerating voltage (5 KeV) and beam current (86 pA) to mitigate charging and minimize degradation of the delicate SEI. The sample preparation was done insight an Ar glovebox. The insulating glass sheet that covered parts of the Cu electrode was cut off and separated from the electrode with the SEI layer (the latter is typically 6 mm long). The SEI containing Cu electrodes were transferred onto SEM specimen stubs with carbon tape and subsequently transported to the SEM room in an Ar filled container. The stubs were shortly exposed to air during the transfer step from the container into the SEM analysis chamber.

The coordination chemistry of the $Li^+$-THF-TFSI$^-$ system was studied with Raman spectroscopy. The Raman measurements were performed on a Horiba Jobin Yvon LabRAM HR800 Raman Spectrometer, using an excitation wavelength of 515 nm (Cobolt Fandango™ 50) and a 50x objective lens. Raman measurements were recorded between 100 and 2000 cm$^{-1}$ using an acquisition time of 30 s and 10 accumulations. We used the maximum slit and hole opening (1000), a 10% neutral density (ND) filter and a grating of 1800. To avoid air and moisture exposure, the electrolyte solutions were prepared in the glovebox and transferred to a pre-dried and air-tight homemade optical sample holder.

### Ammonia quantification

All electrolyte samples were analyzed using ion chromatography (IC, Dionex Aquion from Thermo Scientific) with an autosampler (Dionex AS-AP). The autosampler injects 250 μL aliquots into the 25 μL sample loop, where it is diluted with 2.6 mM methanesulfonic acid (eluent) at a flow rate of 1 mL/min upon injection. The total acquisition time was 10 min. Tubing, connections and the injection needle are made from PEEK, thus being compatible with organic solvents. The IC column (Dionex IonPac CS12A, 4 × 250 mm) is packed with ethylvinylbenzene/divinylbenzene. A guard column (Dionex IonPac CG12A, 4 × 50 mm) was installed upstream to extend the lifetime of the main column. The IC is equipped with an additional electrolytic suppressor (Dionex CDRS 600, 4 mm) to remove conductive ions from the eluent for improving the sensitivity of the conductivity detector. As precaution, the electrolyte samples were diluted with ultrapure water (200 x for 1 M LiTFSI and 400 x for 2 M LiTFSI) to protect the column, which is not compatible with alcohols. To construct the calibration lines (see Supplementary Fig. 31), seven concentrations of $NH_4Cl$ (99.99%, Sigma-Aldrich) in water, 0.002 M LiTFSI (1 M LiTFSI 200 x diluted) and 0.005 M LiTFSI (2 M LiTFSI 400x diluted) were prepared with their respective concentrations of 5, 10, 50, 100, 200, 300, 500 μM. Additional calibration lines in diluted LiTFSI solutions were necessary to compensate for the overlapping $Li^+$ shoulder peak with $NH_4^+$ (Supplementary Fig. 32).

### Data availability

All data supporting the findings of this study are included within the paper and its Supplementary Information. Source data supporting the main figures in the manuscript and the figures found in the Supplementary Information are provided with this paper. Additional data can be obtained from the corresponding author R.K. upon request and may be used for research purposes. Requests for access will typically be addressed within 10 days. Source data are provided with this paper.

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

## Acknowledgements

This research is funded by the Nitrogen Activation and Ammonia Oxidation project within the Electron to Chemical Bonds consortium with project number P17-08, and the Open Technology research program with project no. 15234, which are both financed by The Netherlands Organization for Scientific Research (NWO) and affiliated industrial partners. The authors would like to express their gratitude to S. Ganapathy for her help with the SS NMR measurement.

## Author contributions

B.I. and R.K. conceptualized the paper. B.I. developed the electrochemical testing station, cleaning and drying procedures, and the product quantification. N.G. and B.I. designed the autoclave cell. M.W. and B.I. developed a preparation method for the LFP-reference electrode. B.I. performed, with assistance from A.T., the electrochemical measurements. P.K. carried out the solid-state NMR and XPS experiments, S.P. performed the SEM measurements, and R.W.A.H. executed XRD measurements with Rietveld refinement. F.M.M. assisted with the interpretation of the results. B.I. wrote the original draft, supplementary information and designed all the figures, and P.K., A.T., S.P., N.G., M.W., R.W.A.H, F.M.M., R.K. reviewed and edited the manuscript. R.K. supervised the work.

## Competing interests

The authors declare no competing interests.
