## [Transparent Peer Review file · Nature Communications]

The Effect of Applied Potential on the Li-mediated Nitrogen Reduction Reaction Performance

Corresponding Author: Dr Ruud Kortlever

Version 0:

Reviewer comments:

Reviewer #1

(Remarks to the Author)

In this manuscript, Boaz Izelaar et al. investigated the impact of applied potential on the performance of the Li-NRR with a focus on the properties of the SEI. The authors employed a well-designed experimental approach using a reference electrode based on LiFePO_4 to systematically analyze the correlation between overpotential and FE/NH_3 production rates. Overall, the paper presents a thorough and well-structured discussion, covering both experimental observations and mechanistic interpretations. However, some points require further clarification and improved discussion of results. I recommend minor revisions to address the below points before publication.

(1) In Figure 4a, the current density becomes notably unstable beyond -3.66 V vs. SHE, suggesting potential limitations in the reaction system. It would be beneficial if the authors could provide additional discussion on the possible causes of this instability. Could this indicate that the SEI thickness is fluctuating significantly under high overpotential conditions? If so, how might this affect the assumed correlation between overpotential and SEI thickness?

(2) The manuscript reports a maximum FE of 60%, whereas a previous study (Nature 2022 609, 722-727) under similar test conditions achieved nearly 100% FE. It would be good if the authors could explain the discrepancy in the main text.

(3) "The ^{19}F NMR spectra in Supplementary Figure 10 only shows a TFSI⁻ signal (80 ppm), which suggests that the anion does not decompose." However, if TFSI⁻ did not decompose, would it still be possible to observe Li_2S or Li_2S_x in the XPS data? The S2⁻ in Li_2S is derived from TFSI⁻ degradation (ACS Energy Letters 9, 3790-3795 (2024)). It is suggested that the authors revise this sentence for clarity and accuracy.

(4) The section on the energy efficiency of a batch Li-NRR system is well presented. However, it may be beneficial for the authors to further emphasize the necessity of the hydrogen oxidation reaction (HOR) at the anode. Since THF oxidation is not economically viable and its oxidation products may affect SEI and electrolytes, potentially leading to system instability, a discussion on this aspect would strengthen the analysis (Sci. China Chem. 67, 3510-3514 (2024)).

Reviewer #2

(Remarks to the Author)

This paper reveals a correlation between NH_3 FE and production amount according to potential in Li-NRR. Since the potential is considered as a variable, LFP was used as the reference electrode, and a trend of increasing FE was found as the overpotential increases, and this was proven through XPS, ssNMR, etc. to be related to an increase in inorganic species of SEI. This shows that there is a correlation between overpotential and salt reduction kinetics, and it supports the fact that LiF produced at high potential has low conductivity, so the current density is lowered at high overpotential and the production amount is lowered instead. It is systematically done study and a few points need to be addressed to improve the work.

1. For the analysis of the potential effect, a sufficient comparative analysis of the three samples (Low Ewe, Moderate Ewe, High Ewe) is essential, especially on the analysis of Low Ewe case. Specifically, if the SEI at Ewe is insufficient for ssNMR analysis, a comparison of the three cases under 1 M conditions is necessary. In this case, it is necessary to confirm whether LiEtO is observed.

2. The analysis of Ewe is insufficient, requiring further SEI analysis at Ewe. Conducting long-term experiments and analysis could be a method to ensure sufficient SEI formation.

3. Supplementary Fig. 4: Based on the given rationale, SEI formation is expected to differ between Ar and N_2 conditions. Please explain why starting under Ar conditions and then slowly introducing nitrogen does not result in a significant

difference in Li plating overpotential.

4. Sufficient justification for the thickness labeling is necessary, e.g., assigning SEI thickness from cross-sectional SEM images.

5. In general electrochemistry, overpotential is related to resistance. Please explain the overpotential in relation to resistance and SEI thickness, e.g., using EIS analysis.

6. Figure 9(c): The justification for specifying the solvation structures of SSIP and CIP is needed. For instance, in the case of SSIP, is the possibility of two anions existing in the secondary solvation explicitly excluded?

7. The discussion is primarily based on a 2 M concentration. To verify whether the applied potential concept is applicable across different concentrations, additional analysis and comparison at a 1 M concentration are needed.

Reviewer #3

(Remarks to the Author)

This manuscript proposes the effect of applied potential on the Li-mediated nitrogen reduction reaction performance. The topic is interesting, and certainly consistent with the contents to be proposed to the readers of "Nature Communications".

Overall, I think that this manuscript could be accepted if the Authors will be able to take into account the following major revisions (in terms of bibliographic updates, grammar corrections and content deepening):

- It is a detailed end exhaustive work in which a new parameter, the working electrode (WE) applied potential, is evaluated instead of the current density. Three different values of applied potentials for the Li-NRR are studied. The data are rigorous, and the evaluation is supported with different analysis. The work is surely well-structured and performed; however, it is not clear how much it is extremely innovative, both in its assumptions and final observations. Indeed, regarding the reference electrode, several works from Imperial College already deeply studied it. Could the authors better highlight which is the novelty?

- It is not clear the explanation of the unstable behavior of the current at the lower potential (higher in modulus) in correlation with the observed analysis. The correlation of the SEI layer with salt concentration, and the one of the WE potential with the salt reduction, was reported.

- The preparation of samples for SS-NMR is not clearly explained.

- Testing experiments lasts 4 h, which is a huge duration: the authors must demonstrate that NH₃ does not oxidize in this timeframe.

- Looking at Fig. 10, it does not seem that this work is well placed in the scientific community state of art. Why it should be published in a Nature journal?

- When testing the 3 potential values, the authors study the Li⁺ permeability: it is fine, but the most critical part is the N₂ permeability (which is a more relevant limiting factor). Could you address this point?

- A long paragraph is present on LiF, however it is not corroborated by experimental evidence.

Version 1:

Reviewer comments:

Reviewer #1

(Remarks to the Author)

I appreciate the authors' thorough revisions in response to the reviewers' comments. My concerns have been satisfactorily addressed, and I recommend the manuscript for publication.

Reviewer #2

(Remarks to the Author)

The revision well addressed the issues raised by the reviewer and now is acceptable.

Reviewer #3

(Remarks to the Author)

The manuscript has been properly amended and I recommend its publication.

The Effect of Applied Potential on the Li-mediated Nitrogen Reduction Reaction Performance

Boaz Izelaar, Pranav Karanth, Arash Toghraei, Santosh K. Pal, Nandalal Girichandran, Mark Weijers, Ruud W. A. Hendriks, Fokko M. Mulder, Ruud Kortlever*

REVIEWER REPORTS:

We would like to thank all the reviewers for their positive and constructive feedback. We used it to further strengthen our manuscript, as detailed below.

In blue: Response to the reviewer comments.

In green: Major additions to the manuscript (also highlighted in revised manuscript).

Reviewer: 1

Recommendation: Minor Revision

Comments:

In this manuscript, Boaz Izelaar et al. investigated the impact of applied potential on the performance of the Li-NRR with a focus on the properties of the SEI. The authors employed a well-designed experimental approach using a reference electrode based on Li_xFePO_4 to systematically analyze the correlation between overpotential and FE/NH_3 production rates. Overall, the paper presents a thorough and well-structured discussion, covering both experimental observations and mechanistic interpretations. However, some points require further clarification and improved discussion of results. I recommend minor revisions to address the below points before publication.

(1) In Figure 4a, the current density becomes notably unstable beyond -3.66 V vs. SHE, suggesting potential limitations in the reaction system. It would be beneficial if the authors could provide additional discussion on the possible causes of this instability. Could this indicate that the SEI thickness is fluctuating significantly under high overpotential conditions? If so, how might this affect the assumed correlation between overpotential and SEI thickness?

We thank the reviewer for the feedback and interesting comments. Indeed, a dedicated discussion of the current instabilities observed in Figure 4a was missing in the manuscript and will be included in the rebuttal.

After re-evaluating our results, we indeed expect that the current instabilities are correlated with dynamic growth, breakdown and SEI morphology changes during operation. Since Li^+ plating is considered to be the main electrochemical reaction at the working electrode, the current is directly proportional to the Li^+ flux through the SEI. Hence, the observed instability in the current is likely correlated with dynamic growth, breakdown and SEI morphology changes during operation at these potentials. At -3.2 V, the current is relatively stable and can be related to a relative thin and porous

SEI (Figure 5a-c and Supplementary Fig. 11-12) that exerts a relative low Li^+ transport resistance. The chronoamperometry measurements below -3.7 V start to show current response instabilities, and correlates well with thicker and denser SEI morphologies at more negative applied potentials (Figure 5d-i and Supplementary Fig. 13-16).

To further address the response of comment 1 we added the following to section "Potential effect on the composition and morphology of the SEI" last paragraph:

The current response is a direct measure of the Li plating rate and the Li^+ flux as Li plating is considered to be the only electrochemical reaction occurring at the working electrode. Therefore, instabilities in the current response (Fig. 4a) are likely related to a dynamic process between SEI thickening and breakdown, leading eventually to an increase or decrease of the Li^+ transport resistance during chronoamperometry measurements.

Fig. 4 | Potential dependency of the Li-mediated nitrogen reduction performance. a Chronoamperometry measurements using 2 M LiTFSI (with 0.1 M EtOH) as electrolyte.

Fig. 5 | SEM images of the Cu electrodes post-measurement. An overview of the SEM images of the Cu electrodes showing the SEI morphology and the Cu-SEI interface. These electrodes were obtained after 4 hour chronoamperometry experiments at -3.2 V with different magnifications: **a** 250x, **b** 2000x, **c** 20000x. At -3.7 V with different magnifications: **d** 250x, **e** 2000x, **f** 20000x. At -4.6 V with different magnifications: **g** 250x, **h** 2000x, **i** 20000x. Additional SEM images of the SEI's are available in Supplementary Figures 11-13. 2 M LiTFSI in 0.1 M EtOH/THF was used as the electrolyte.

Supplementary Fig. 11. SEM images of the Cu electrode post-measurement at -3.2 V, focusing on the SEI. (a,b) have a magnification of 5000x and (c) 12000x. The electron beam-induced charging effects are related to the insulating properties of the SEI. The SEI thickness varies between 1-7 μm and can only be used as a rough estimate because the cross-section is poorly defined, and the thickness varies along the length of the SEI. The thickness labels in the figure were obtained by taking the average and the standard deviation of 8 different spots along the dotted line. The length scales were obtained with ImageJ.

Supplementary Fig. 12. Additional scanning electron microscopy images of the Cu electrode post-measurement at -3.2 V, focusing on the SEI. (a,b) have a magnification of 5000x and (c) 25000x. The electron beam-induced charging effects are related to the insulating properties of the SEI.

Supplementary Fig. 13. Additional scanning electron microscopy images of the Cu electrode post-measurement at -3.7 V, focusing on the SEI at different magnifications. (a) 150x, (b) 2000x, (c) 20000x and 50000x for the inset. Spots with a darker contrast in (c) may indicate nanosized pores in the SEI structure.

Supplementary Fig. 14. SEM image of the Cu electrode post-measurement at -3.7 V, focusing on the SEI cross-section at 100 x magnification. The SEI thickness varies between 0.8-1 mm, which can be only be used as a rough estimate because the cross-section is poorly defined, and the thickness varies along the length of the Cu wire. The thickness labels in the Figure were obtained by taking the average and the standard deviation of 8 different spots along the dotted line. The length scales were obtained with ImageJ.

Supplementary Fig. 15. Additional scanning electron microscopy images of the Cu electrode post-measurement at -4.6 V, focusing on the SEI at different magnifications. (a) 200x, (b) 1000x, (c) 10000x.

High E_{we}
(-4.6 V vs. SHE)

Supplementary Fig. 16. (a) SEM image of the Cu electrode post-measurement at -4.6 V, focusing on the SEI thickness at 150 x magnification. The indicated SEI thickness of $\sim 300 \mu\text{m}$ is a rough estimate because the cross-section is poorly defined, and the thickness varies along the length of the Cu wire. (b) SEM image of another SEI covering the Cu electrode at 100 x magnification without cross-section. The thickness of the Cu wire including SEI is $\sim 1 \text{ mm}$ (measured with ImageJ). By subtracting the Cu wire thickness (0.5 mm) and dividing it by two, the SEI thickness becomes roughly $250 \mu\text{m}$, which is in the same ballpark range as the estimated SEI thickness in (a).

(2) The manuscript reports a maximum FE of 60%, whereas a previous study (*Nature* 2022 609, 722-727) under similar test conditions achieved nearly 100% FE. It would be good if the authors could explain the discrepancy in the main text.

We agree with the reviewer that there is indeed a discrepancy between our work and the reference paper. Although the main intention of our work is to focus more on the fundamental aspects of the Li-NRR, it is indeed important for the reader to understand why our system shows a lower FE with respect to the previous reference study.

The authors from *Nature*, 2022, 609, 722-727 noticed that the FE_{NH_3} increased with the experimental measurement time and became more or less constant after 6 hour (Fig. R1). They assume that especially during the first three hours of measurement, a significant amount of charge is lost due to the build-up of a SEI layer. Unfortunately, we could only perform 4 hour experiments due to time constraints, and again, maximizing the performance is not our main intention. Remarkably, by performing a linear interpolation between 3 and 6 hours of their time dependent data (Figure 1), the FE is 72% at 4 hours, which is within a reasonable range of our obtained FE (62% at -4.6 V).

Other factors, such as salt purity, hydrodynamics within the cell (stirring rate and design), electrode configuration, were earlier reported to influence the Li-NRR performance [*Nature*, 2022, 609, 722-727; *J. Mater. Chem. A*, 2023, 11, 12746; *Joule*, 2019, 3, 1127–1139; *J. Phys. Chem. C* 2021, 125, 11402–11410].

Paraphrased from *Nature* 2022 609, 722-727 (on page 725):

“Faradaic efficiency at 3 h was $65 \pm 8\%$, whereas experiments undertaken for 6 h, 9 h and 12 h produced ammonia at progressively increasing faradaic efficiencies of $85 \pm 9\%$, $89 \pm 9\%$ and $91 \pm 7\%$, respectively; these values are based on the charges and NH_3 yields over the whole duration of the experiments, including the initial period of lower productivity. Calculation of the faradaic efficiency for the 6–12 h experiments after subtraction of the charge and NH_3 yield data for the 3 h tests produced values very close to 100% (empty triangles in Fig. R1/Fig. 3b). In other words, essentially all charge passing through the system after 3 h was associated with the conversion of N_2 to NH_3 and the excess charge during the first 3 h was probably associated with the electrolyte-solution decomposition, which stops contributing after 3 h.”

To further address the response of comment 2 we added the following to section “Energy efficiency of a batch Li-NRR system”:

MacFarlane, Simonov and coworkers managed to obtain an EE of 13% when excluding the SEI formation period, as they reached a FE near unity over longer testing periods while using similar operating conditions (2 M LiTFSI in 0.1 M EtOH/THF with 15 bar N_2 pressure).⁶ We did not intend to optimize for Li-NRR performance metrics, but reach a similar FE over shorter tests where ammonia synthesis and SEI formation occur simultaneously, while remaining discrepancies may be allocated to the salt purity,^{11, 19} hydrodynamics of the cell,⁶ and electrode configuration.⁵⁶

Figure R1. Modified and taken from *Nature* 2022 609, 722-727 (Fig. 3b). Linear interpolation between 3- and 6-hour data point of the Faradaic efficiency. The star indicates the interpolated FE_{NH_3} after 4 hrs in the particular publication.

(3) “The ^{19}F NMR spectra in Supplementary Figure 10 only shows a TFSI⁻ signal (80 ppm), which suggests that the anion does not decompose.” However, if TFSI⁻ did not decompose, would it still be possible to observe Li_2S or Li_2S_x in the XPS data? The S2⁻ in Li_2S is derived from TFSI⁻ degradation (*ACS Energy Letters* 9, 3790-3795 (2024)). It is suggested that the authors revise this sentence for clarity and accuracy.

The paragraph in question discusses the post-measurement liquid NMR results from the electrolyte to trace back any oxidation reactions that occurred in our Li-NRR system. We agree with the reviewer that we cannot entirely exclude TFSI decomposition at the anode due to the detection limit of the ^{19}F NMR measurement. However, according to the battery literature TFSI decomposition is unlikely because etheral solvents have a significantly lower anodic stability and tend to oxidize at 1 V vs. SHE because they do not possess electron-withdrawing groups [*Woodhead Publishing*, 2011, 484-515; *J. Am. Chem. Soc.* 2011, 133, 13121–13129; *Current Opinion in Electrochemistry*, 2019,13, 86–93]. Moreover, other Li-NRR literature reports state similar conclusions, wherein their oxidation products originate from etheral solvent and ethanol [*Energy Environ. Sci.*, 2023,16, 1082-1090; *ChemSusChem* 2023, e202301011; *RSC Adv.*, 2021, 11, 31487]

To further address the response of comment 3 we added the following to section “Relationship between the potential and the Li-NRR performance”:

The ^{19}F NMR spectra (Supplementary Fig. 10) only shows a TFSI⁻ signal (80 ppm), suggesting that the anion does not oxidize during an electrochemical measurement, although we cannot exclude TFSI⁻ oxidation completely due to the detection limit. An elaborate analysis of the liquid NMR results can be found in Section S1. of the Supplementary Discussion.

(4) The section on the energy efficiency of a batch Li-NRR system is well presented. However, it may be beneficial for the authors to further emphasize the necessity of the hydrogen oxidation reaction (HOR) at the anode. Since THF oxidation is not economically viable and its oxidation products may affect SEI and electrolytes, potentially leading to system instability, a discussion on this aspect would strengthen the analysis (*Sci. China Chem.* 67, 3510-3514 (2024)).

We agree with the comments from the reviewer and included this in the rebuttal.

To further address the response of comment 4 we added the following to section “Energy efficiency of a batch Li-NRR system”:

Additionally, the side products of THF oxidation may compromise the properties of the electrolyte and the SEI, leading potentially to system instabilities. According to our liquid NMR results, EtOH is also consumed by these side products, indicating that sufficient proton concentrations cannot be sustained for long term measurements. Hence, a Li-NRR system based on THF oxidation is most likely not technically and economically feasible, and an H source from H₂ oxidation would be preferred.¹

Reviewer: 2

Recommendation: Major Revision

Comments:

This paper reveals a correlation between NH₃ FE and production amount according to potential in Li-NRR. Since the potential is considered as a variable, LFP was used as the reference electrode, and a trend of increasing FE was found as the overpotential increases, and this was proven through XPS, ssNMR, etc. to be related to an increase in inorganic species of SEI. This shows that there is a correlation between overpotential and salt reduction kinetics, and it supports the fact that LiF produced at high potential has low conductivity, so the current density is lowered at high overpotential and the production amount is lowered instead. It is systematically done study and a few points need to be addressed to improve the work.

1. For the analysis of the potential effect, a sufficient comparative analysis of the three samples (Low Ewe, Moderate Ewe, High Ewe) is essential, especially on the analysis of Low Ewe case. Specifically, if the SEI at Ewe is insufficient for ssNMR analysis, a comparison of the three cases under 1 M conditions is necessary. In this case, it is necessary to confirm whether LiEtO is observed.

We thank the reviewer for the feedback and interesting comments. It is indeed unfortunate that we were not able to perform ssNMR in the *low E_{we} regime*. However, we are convinced that the XPS depth profiling and SEM results (Fig. 6, Fig. 5a-c) are sufficient to conclude LiEtO is the main SEI species at low overpotentials. Additionally, our XPS results and SEM images match very well with *Nature Energy* 2022, 8, 138-148, wherein LiEtO was identified as the main SEI constituent.

The main aim of our paper is to investigate the relationship between the applied potential and the Li-NRR performance. For the majority of the analysis, we used 2 M LiTFSI because this was reported earlier as a high-performance electrolyte [*Nature*, 2022, 609, 722-727]. To in- or exclude any possible correlations between the electrolyte concentration and the applied potential, we decided to include 1 M LiTFSI as well. Unfortunately, this was not clearly stated in the manuscript and is made more explicit in the rebuttal.

We noticed a significant reduction in both the FE_{NH₃} and R_{NH₃} when 1 M LiTFSI was used, while the respective trends as a function of the applied potential remain similar to what was observed with 2 M LiTFSI. Therefore, the loss of Li-NRR performance at lower electrolyte concentrations is likely related to differences in the Li-ion solvation environment. While this has already been extensively discussed in *J. Mater. Chem. A*, 2023, 11, the solvation environment of LiTFSI in the context of Li-NRR has only been studied to a limited extend. The Raman spectra in combination with the XPS data at 1 M and 2 M LiTFSI supports our conclusions for the electrolyte concentration effects.

To further address the response of comment 1 we added the following to section "Electrochemical characterization of the Li-NRR system":

The main part of the analysis will be done with 2 M LiTFSI dissolved in 0.1 M EtOH/THF as this electrolyte was earlier identified by Du et al. as a high-performance electrolyte.⁶ Additionally, we performed Li-NRR experiments with 1 M LiTFSI to analyse any possible correlations between the electrolyte concentration and the applied potential.

To further address the response of comment 1 we added the following to section “Electrolyte concentration effects”:

LiTFSI concentration effects in the context of Li-NRR performance have only been studied to a limited extent. To that end, we used Raman spectroscopy to study the coordination chemistry of the Li^+ - TFSI $^-$ - THF solvation environment in the bulk electrolyte. The SEI composition was also analysed with XPS, wherein we focused particularly on the *high E_{we} regime* to compare the LiF concentrations between 1 M and 2 M LiTFSI.

Fig. 6 | XPS depth profiling of the solid electrolyte interphases. a Elemental composition of the SEI obtained post-measurement at -3.2 V, -3.7 V and -4.6 V after 0-600 s of Ar^+ etching. **b** High resolution XPS spectra of C 1s, F 1s, S 2p and O 1s of the SEI obtained post-measurement at -3.2 V, -3.7 V and -4.6 V after 0 s, 60 s, 240 s and 600 s of Ar^+ etching. Vertical dashed lines indicate binding energies of known chemical species. Reference lines in the S 2p spectra indicates the binding energy of the S $2p_{3/2}$ orbital. Surface scan of LiTFSI is also included for referencing. Corresponding XPS survey spectra are available in Supplementary Figures 18-20. 2 M LiTFSI in 0.1 M EtOH/THF was used as the electrolyte.

2. The analysis of Ewe is insufficient, requiring further SEI analysis at Ewe. Conducting long-term experiments and analysis could be a method to ensure sufficient SEI formation.

The clarity of the writing discussing the relationship between E_{we} , SEI and Li-NRR performance in section “Potential effect on the composition and morphology of the SEI” was indeed insufficient and has been significantly revised in the rebuttal (also in accordance with comments from reviewer 3). We are convinced that our discussion is already supported by sufficient experimental data. Hence, we deem performing more experiments is not necessary.

To further address the response of comment 2 we updated the entire section “Potential effect on the composition and morphology of the SEI” (unchanged text is indicated in black):

Potential effect on the composition and morphology of the SEI

The equilibrium potential of Li plating is considered to be more negative than the reduction potential of THF (+0.50 V vs Li/Li⁺),³⁶ TFSI⁻ (+0.75 V vs Li/Li⁺),³⁷ and EtOH (+1.00 V vs Li/Li⁺).³⁸ Therefore, the kinetic stability of the reactants increases in order of THF > TFSI⁻ > EtOH, indicating that there can be a correlation between the SEI composition and the electrode potential. To further understand the earlier established relationship between the potential and the Li-NRR performance with 2 M LiTFSI, we carried out post-measurement characterization with SEM, XPS and ssNMR to analyse the morphology and composition of the SEI within the earlier defined potential regimes. To ensure that the SEI layer remains in-tact, we always removed the electrolyte with the purge valve before degassing the autoclave cell (Supplementary Fig. 1b). The retrieved electrodes were washed with THF to remove residual salts and subsequently dried in the glovebox.

At -3.2 V (*low* E_{we} regime), the SEM images (Figs. 5a and 5b) of the Cu wire reveal a thin cracked layer, which can be identified as the SEI. These cracks are most likely the result of contraction during THF evaporation. *The curvature of the wire and the cracks reveals cross-sectional views of the SEI which can be used as a qualitative indicator of the SEI thickness and reveal morphological information (see Supplementary Fig. 11). The SEI thickness varied over the length of the Cu wire and is estimated to be between 1-7 μm .* The surface under the SEI (Fig. 5c) highlights nano-spherical deposits, which extends into a chain linked macro-porous network (Supplementary Fig. 12) until the layer becomes uniformly passivated, showing a relatively smooth surface. The XPS surface scan indicates a strong carbon signal (Fig. 6a), with a prominent C-C (284.8 eV) peak in the high-resolution C 1s spectra (Fig. 6b), which is often associated with adventitious carbon. The XPS depth profile analysis reveals an elemental shift towards organic species with shorter carbon-chains and a higher oxygen content, indicating a more ethanol-derived layer. Besides Li ethoxide as being reported as the main product of EtOH, we also observe a C=O peak in the C 1s spectra, which is typically affiliated with Li carbonate and can suggest an alternative decomposition pathway. This agrees well with the O 1s spectra, where the singlet peak could originate from C-O (532.7 eV), C=O (531.5 eV) and Li alkoxide (530.4 eV) signal contributions.^{30,39} Our surface structure and composition shows similarities with the work of Steinberg et al.,³⁰ wherein they did not identify a metallic Li phase, but mostly a poor passivation layer of ethanol-derived species.

Based on the earlier mentioned equilibrium potentials, we expected that all species in the electrolyte would decompose at -3.2 V and form a mixture of both organic and inorganic SEI species, while the XPS results clearly indicate an EtOH-derived layer. We propose that the EtOH decomposition reaction on Li⁰ is preferred because it is the most kinetically unstable compound in the electrolyte. Consequently, the thin Li-alkoxy passivation layer formed upon electrode polarisation can hinder the decomposition reaction of other species in the electrolyte.³⁶ Although ammonia production was observed, the low FE_{NH_3} (< 20%) indicates that nitrogen activation is not favourable at these conditions,

which is most likely related to an imbalance in the transport rates of Li^+ , H^+ , and N_2 diffusion through the SEI layer. Since our operating conditions (N_2 pressure, EtOH and salt concentration) are relatively similar to previous literature reports observing higher R_{NH_3} and FE_{NH_3} 's, we expect that the properties of the SEI at -3.2 V is mainly responsible for the low FE_{NH_3} . Additionally, Spry *et al.* and Benedek *et al.* operated at ambient N_2 pressures and managed to obtain similar FE_{NH_3} 's (~20%),⁴⁰ and even higher (40%).⁴¹ Hence, increasing the N_2 operating pressure does not directly result in better Li-NRR performance. In partial agreement with the work of Chorkendorff, Norskov and coworkers,²⁰ it seems that the SEI properties have a much stronger effect on regulating the transport rate of Li^+ and H^+ than on N_2 . The porous Li-ethoxide structure seems to be particularly poor in slowing down the Li^+ and H^+ diffusion rate, leading to build-up of more SEI material (due to rapid and uncontrolled Li plating) and the formation of hydrogen gas. The latter explains the visible macro pores and cavities in the internal segment (Supplementary Fig. 12b) and top surface of the SEI (Supplementary Fig. 12c).

At more negative E_{we} , the SEI layers were hundreds of micrometres thick and difficult to analyse with SEM. Therefore, we decided to break the layer in a controlled manner until the Cu wire and a cross-sectional view of the deposits were exposed (Fig. 5d-i, Supplementary Fig. 13-16). This allowed us to analyse the Cu interface and the internal morphology of the SEI but did not result in a well-defined cross-sectional surface, leading to rougher estimates of the SEI thickness. At -3.7 V, the Li microstructure on the Cu interface changes from particle-like to dendritic features (Fig. 5f), which can be signs of a diffusion limited growth regime.⁴² The Cu interface at -4.6 V also contains Li deposits, but with a more rod-like dendritic geometry surrounded by a dense SEI layer (Fig. 5i). The elemental composition of the SEI at -3.7 V and -4.6 V (Fig. 6a) reveals an organic surface layer (most likely adventitious carbon), while the subsurface layers are predominantly inorganic with an increasing order of $\text{Li} > \text{F} > \text{C} > \text{O} > \text{S} > \text{N}$ present. The F 1s spectra in Figure 6b discloses a prominent LiF peak (684.5 eV), wherein the majority of elemental fluorine is in the form of LiF via LiTFSI decomposition,⁴³ which is in agreement with other studies employing a F-based salt.^{6, 20} Other LiTFSI decomposition products via its sulfone groups, such as Li_2SO_4 (166.8 eV) and a mix of Li_2S_x (Li_2S_6 at 162.8 eV, Li_2S_4 at 161.2 eV) and Li_2S (159.8 eV) were also identified in the S 2p spectra (given binding energies are from the S $2p_{3/2}$ orbital),^{43, 44} but remained in low quantities. Deconvolution of the Li 1s peak is challenging because it resembles a singlet representing all Li species with overlapping binding energies (Supplementary Fig. 17). In this work, ssNMR is used as a complementary characterization technique and is especially useful for the identification of several SEI materials in the bulk phase, such as metallic Li, LiTFSI, LiF and organic species based on their unique chemical shift. Unfortunately, the application of ssNMR was unsuccessful at -3.2 V due to the limited availability of SEI material.

At -3.7 V, the absence of a metallic Li peak in the ^7Li NMR spectra (Fig. 7a) indicates that the layer of Li dendrites (observed by SEM), is thin and only present on the Cu interface. Based on ^{19}F NMR, LiF (-203 ppm) represents most of the fluorine compounds in the SEI (55±8%) and agrees well with our XPS results (see Supplementary Table 2). When shifting to more negative potentials (-4.6 V), the LiF content (74%) with respect to LiTFSI (26%) increases substantially, suggesting TFSI⁻ reduction requires a significantly higher activation barrier than the solvent decomposition reactions. Additionally, a metallic Li peak (at 265 ppm) becomes visible in the ^7Li NMR spectra (Fig. 7a), indicating that the freshly electroplated Li^0 does not immediately react with the electrolyte or solvent species as occurs at low E_{we} . Higher concentrations of LiF seems to correlate well with the existence of a metallic Li^0 layer, and agrees with the notion that LiF has better electron insulating properties than other SEI constituents. Despite the majority of the SEI being inorganic, THF-based species are also revealed in XPS C 1s spectra (Fig. 6), ^1H NMR (Fig. 7c) and ^{13}C NMR (Supplementary Fig. 21) spectra. Ethoxide-species were not detectable, suggesting that its concentration is below the limit of detection or short-lived and immediately re-dissolve back into the electrolyte to act as a proton shuttle as was previously

suggested.^{16, 30, 46} Dissolution of Li ethoxide is further supported by the observation of a two orders of magnitude thinner SEI at -3.2 V (~1-7 μm) in comparison with the substantial inorganic layers at -3.7 V (~0.8-1 mm) and -4.6 V (~0.25-0.30 mm, Supplementary Fig. 11, 14 and 16), while the accumulated charge differs only by a factor of 6.9 and 3.3, respectively (Supplementary Fig. 7).

We find a clear correlation between the FE_{NH_3} and the LiF concentration in the SEI (induced by the potential driving force). LiF is not necessary to make ammonia, but simulations based on first principles have pointed out that the Li^+ transport resistance within LiF layers is much higher in comparison to other SEI species.²⁰ Therefore, the Li^+ diffusion rate (and perhaps also the H^+ diffusion rate as was pointed out by us earlier) slows down relative to N_2 transport and this increases the lithium nitridation probability, leading ultimately to higher FE_{NH_3} . These findings match well with our experimental observations, but the structure of our SEI is more complicated and most likely resembles a mixture of polyhetero organic and inorganic microphases that is typically observed in Li-ion batteries.¹⁷ Hence, we cannot exclude the influence of other species than LiF entirely.

Thicker and especially denser SEI structures obtained at -3.7 V and -4.6 V (in comparison with -3.2 V) seem to also correlate with the FE_{NH_3} . A similar observation was earlier reported by McShane et al., wherein the FE_{NH_3} monotonically increases with the SEI thickness. We expect that the more tortuous paths from the bulk towards the electrode surface slow down the Li^+ (and probably H^+) transport rate, while N_2 is less affected. The exact mechanism at play remains unknown and requires further investigation. The current response is a direct measure of the Li plating rate and the Li^+ flux because Li plating is considered to be the only electrochemical reaction at the working electrode. Therefore, instabilities in the current response (Fig. 4a) could be related to a dynamic process between SEI thickening and breakdown, leading eventually into an increase or decrease of the Li^+ transport resistance during chronoamperometry measurements.

Fig. 7 | Solid-state NMR spectra of the solid electrolyte interphases. The SEIs for ssNMR were obtained after chronoamperometry at -3.7 V and -4.6 V. **a** ⁷Li NMR spectra with signals at 0 ppm and -265 ppm indicate LiTFSI, Li-SEI materials and metallic Li⁰, respectively.¹⁷ **b** ¹⁹F NMR spectra with a large peak at -80 ppm, representing a -CF₃ contribution from LiTFSI or derivative products. The small and broad peak at -203 ppm is attributed to LiF. **c** ¹H NMR spectra have two broad peaks at 3.63 ppm and 1.75 ppm that match with THF. The SEI compound distribution is summarized in Supplementary Table 2. 2 M LiTFSI in 0.1 M EtOH/THF was used as the electrolyte.

Fig. 8 | Schematic of the SEI composition, thickness and Li morphology at different applied potentials. At -3.2 V, the SEI was only a few μm thick, showing particle-like Li microstructure and a mostly organic composition. At -3.7, the SEI grew into a layer of hundreds of micrometer thick, consisting mostly of inorganic F-species, while the morphology of the Li deposits is mostly dendritic. At -4.6 V, the Li microstructure is also dendritic, but with a more rod-like geometry surrounded by a dense LiF-enriched SEI. The patch work of different SEI microphases resembles the mosaic pattern that is typically observed in Li-ion batteries and also in the Li-NRR context.^{17, 30}

3. *Supplementary Fig. 4: Based on the given rationale, SEI formation is expected to differ between Ar and N₂ conditions. Please explain why starting under Ar conditions and then slowly introducing nitrogen does not result in a significant difference in Li plating overpotential.*

It is important to mention that the referred CVs (Supplementary Fig. 4) by the reviewer were obtained in an EtOH-containing electrolyte, meaning that the SEI formation during Li cycling will most likely have a EtOH-based SEI layer under both Ar and N₂ atmosphere. This could be the reason for small changes in the Li plating overpotential. However, the Li plating and stripping current-voltage relationship becomes very distinct when N₂ is introduced into the cell's headspace. This could indicate that N₂ does influence the properties of the SEI via the formation of Li_xN_y or Li_xN_yH_z species.

Supplementary Fig. 4. Li⁺ reduction peaks in the cyclic voltammograms are changing when releasing 2.5 bar N₂ pressure into the autoclave cell after the first scan. The first scan is a voltammetry under Ar which is comparable with Figure 2c from the main manuscript. The scan rate is 20 mV s⁻¹ and the electrolyte is 2 M LiTFSI in 0.1 M EtOH/THF. The potential is for 85% IR_u compensated by the build-in software in EC-Lab.

4. *Sufficient justification for the thickness labeling is necessary, e.g., assigning SEI thickness from cross-sectional SEM images.*

We thank the reviewer for this comment. We agree with the reviewer that further justification of the allocated SEI thicknesses is required. After (re)-analyzing SEM images with ImageJ, we noticed thickness variations along the area and length of the Cu electrode, which makes it difficult to estimate a general layer thickness. We decided to allocate an approximate range for the SEI thicknesses because the amount of SEM images we have with SEI cross-sections is small. For -3.2 V, the thickness varied between 1-7 μm , while they increase by almost three orders of magnitude at -3.7 V (approx. 800-1000 μm) and two orders of magnitude higher at -4.6 V (approx. 250-300 μm). Although it was not our aim to perform a quantitative SEI thickness study, these rough estimate thicknesses are still useful for relative comparison purposes and give an idea of the length scales.

Additional SEM images were added to the Supplementary Information to support these estimates (Supplementary Fig. 11, 14 and 16). We refer the reviewer to our response to comment 1 of reviewer 1 for the SEM images.

To further address the response of comment 4 we added the following to paragraph 2 of section “Potential effect on the composition and morphology of the SEI”

The curvature of the wire and the cracks reveals cross-sectional views of the SEI which can be used as a qualitative indicator of the SEI thickness and reveal morphological information (see Supplementary Fig. 11). The SEI thickness varied over the length of the Cu wire and is estimated to be between 1-7 μm .

To further address the response of comment 4 we added the following to paragraph 4 of section “Potential effect on the composition and morphology of the SEI”

This allowed us to analyse the Cu interface and the internal morphology of the SEI but did not result in a well-defined cross-sectional surface, leading to rough estimates of the SEI thickness.

5. In general electrochemistry, overpotential is related to resistance. Please explain the overpotential in relation to resistance and SEI thickness, e.g., using EIS analysis.

We thank the reviewer for this suggestion and updated the manuscript accordingly. The write-up in the energy efficiency section is now significantly improved by relating the overpotential to the resistances in the cell, such as the Li^+ transport resistance in the SEI, Li^+ charge transfer resistance and cathodic and anodic charge transfer resistances. We also noticed a missing Nyquist plot that should have supported the value for the ohmic loss in the cell (71 Ω) and has been added to the supplemental information.

To further address the response of comment 5 we added the following to section “Energy efficiency of a batch Li-NRR system”:

The voltage losses in our cell are related to the electrode overpotentials and ohmic losses via resistive dissipation (Fig. 10), wherein the ohmic contributions only become significant at high current densities due to our compact cell design ($R_{\Omega} = 71 \Omega$, Supplementary Fig. 26) and small working electrode area (0.1 cm^2). The electrode overpotentials are predominantly associated with the Li^+ transport resistance through the SEI and THF oxidation, where the former becomes more significant in the high E_{we} regimes due to the build-up of a thicker and denser SEIs. Based on the cyclic voltammograms, the Li^+ charge transfer resistance cannot be fully excluded ($\sim 0.3 \text{ V}$ at -45 mA cm^{-2} , Fig. 3d) but contributes to a lesser extent. Anodic overpotentials are related to Pt surface poisoning by organic residues from solvent oxidation, which increase the charge transfer resistance, but can be reduced by substituting the sacrificial solvent with hydrogen oxidation as a proton source.

Supplementary Fig. 26. Nyquist plot of the autoclave cell in a two-electrode configuration with Cu and Pt as a working and counter electrode, respectively. 2 M LiTFSI in 0.1 M EtOH/THF was used as the electrolyte. The ohmic resistance (R_{Ω}) is the impedance contribution at high EIS frequencies and is solely related to the Real Z contribution. R_{Ω} can be obtained from the Nyquist plot by taking the graph's intersection with the Real Z axes, which is equal to 71 Ω .

Fig. 10 | Cell voltage contributions at different applied working potentials. Equilibrium potentials of Li⁺ reduction and THF oxidation are taken as $E_{eq,Li/Li^+} = -3.02$ V and $E_{eq,THF} = +1$ V, respectively. The ohmic overpotential is calculated *via* $\eta_{ohmic} = IR_{\Omega}$ ($R_{\Omega} = 71$ Ω) and the electrode overpotential is the remaining voltage contribution after subtraction ($\eta_{elect} = E_{cell} - E_{eq} - \eta_{ohmic}$). Data is based on the chronoamperometry measurements in Supplementary Figure 7.

6. Figure 9(c): The justification for specifying the solvation structures of SSIP and CIP is needed. For instance, in the case of SSIP, is the possibility of two anions existing in the secondary solvation explicitly excluded?

The solvation structures in Figure 9c indeed need further justification. The schematic is modified towards a more common representation of the Li^+ coordination modes [Nano Energy, 2023, 115, 108722; ACS Energy Lett. 2024, 9, 4883–4891; Colloids Surf. A Physicochem. Eng. Asp. 2023; 674:131831], while the coordination number of Li^+ in non-aqueous solvents can vary between 4-6 [J. Phys. Chem. B 2014, 118, 3689–3695]. For simplification, we assume that Li^+ has four coordinated sites. Aggregated coordination modes were excluded from the schematic because they are not observed in the Raman spectra. A more detailed representation of the LiTFSI solvent coordination structure can be found in the work of Henderson et al. [Phys. Chem. B 2014, 118, 13601–13608].

Fig. 9 | Raman spectroscopy of different LiTFSI concentrations dissolved in THF. **a** Raman shift of the S-N-S bending vibration in crystalline LiTFSI (746.8 cm^{-1}) and TFSI⁻ dissolved in THF between 0.5-2.5 M. The Raman peaks were deconvoluted into a SSIP peak (740.5 cm^{-1}) and CIP peak (745.5 cm^{-1}) in OriginPro 10. **b** Percentage between the SSIP and CIP coordination modes with the LiTFSI salt concentration. **c** Schematic of the SSIP and CIP solvation environment of the Li^+ - THF - TFSI⁻ system based on ref ⁵³.

7. The discussion is primarily based on a 2 M concentration. To verify whether the applied potential concept is applicable across different concentrations, additional analysis and comparison at a 1 M concentration are needed.

We refer the reviewer to our response on comment 1 on page 11 of the rebuttal report, since the nature of this comment is relatively similar.

Reviewer: 3

Recommendation: Major Revision

Comments:

This manuscript proposes the effect of applied potential on the Li-mediated nitrogen reduction reaction performance. The topic is interesting, and certainly consistent with the contents to be proposed to the readers of "Nature Communications". Overall, I think that this manuscript could be accepted if the Authors will be able to take into account the following major revisions (in terms of bibliographic updates, grammar corrections and content deepening):

1. It is a detailed and exhaustive work in which a new parameter, the working electrode (WE) applied potential, is evaluated instead of the current density. Three different values of applied potentials for the Li-NRR are studied. The data are rigorous, and the evaluation is supported with different analysis. The work is surely well-structured and performed; however, it is not clear how much it is extremely innovative, both in its assumptions and final observations. Indeed, regarding the reference electrode, several works from Imperial College already deeply studied it. Could the authors better highlight which is the novelty?

We thank the reviewer for the positive comments. We are indeed aware of the work performed at Imperial College and Stanford who both identified partially delithiated LFP as reliable reference electrode (LFP-RE) for Li-NRR systems in parallel.^{1,2} Their work rather focuses on the properties of the LFP-RE in comparison with other quasi reference electrodes (such as Ag and Pt wires) and discusses its applications for non-aqueous electrochemistry in general. We acknowledged their work in our manuscript and were inspired to adopt it into our high pressure cell.

To the best of our knowledge, we are not aware of any similar works that established a clear relationship between the applied potential (using a reliable RE) and the Li-NRR performance indicators, such as stability, ammonia faradaic efficiency and production rate. Additionally, we performed post-measurement characterization of the electrode interface at low, moderate and high overpotentials in order to get a better understanding of any potentially induced correlations. SEI characterization with solid-state nuclear magnetic resonance spectroscopy (ssNMR) was also performed for the first time in the Li-NRR field and gave unique insights into the composition of the SEI.

To further address the response of comment 1 we added the following to the Introduction (unchanged text is indicated in black):

Herein, we for the first time implement a reliable LFP based RE in a high performance autoclave cell implementing a fluorine-based electrolyte to investigate the relationship between the applied potential and the Li-NRR performance indicators, such as the R_{NH_3} , FE_{NH_3} and reaction stability. This allows us to identify the individual voltage contributions of a batch-type Li-NRR system and use these to optimize the energy efficiency. To probe any potential induced effects on the SEI composition and morphology, we perform post-measurement characterization techniques including X-ray photoelectron spectroscopy (XPS), scanning electron microscopy (SEM) and solid-state nuclear magnetic resonance spectroscopy (ssNMR), wherein the latter has never been implemented in the context of Li-NRR. Electrolyte concentration effects are also included in this study to analyse any existing correlations between the electrolyte concentrations and the potential.

We identify three potential regimes, in which the current response up to -3.2 V (all reported potentials are in V vs SHE) is the most stable, but at the cost of a relatively low FE_{NH_3} (<22%) and R_{NH_3} (<16 nmol

$\text{s}^{-1} \text{cm}^{-2}$). At more negative potentials (down to -4.0 V), both the FE_{NH_3} and R_{NH_3} increases to approximately 50% and $350 \text{ nmol s}^{-1} \text{cm}^{-2}$, respectively. Beyond -4.0 V, breakdown of the current response is initiated, while the FE_{NH_3} reaches to a maximum of $62.9 \pm 2.2\%$ at -4.6 V. SEI characterization results show higher LiF concentrations at more negative potentials, indicating that a significant overpotential is required to overcome the fluorine-based electrode decomposition barrier. The strongest positive correlation was observed between the FE_{NH_3} and the LiF concentration. Thicker and denser SEI morphologies observed at -3.7 V and -4.6 V are also beneficial for the FE_{NH_3} , while they can be responsible for the observed current instabilities beyond -4.0 V. These findings improve the current understanding of the SEI formation process and sheds light on a new optimization strategy for Li-NRR systems, which contribute to the development of a sustainable ammonia production process.

To further address the response of comment 1 we added the following to the conclusion (unchanged text is indicated in black):

In summary, Li-NRR experiments under 20 bar N_2 pressure were for the first time performed with a reliable reference electrode based on a partially delithiated sheet of Li_xFePO_4 . This allowed us to investigate the relationship between the applied potential and the Li-NRR performance indicators, such as the FE_{NH_3} , R_{NH_3} and reaction stability. Additionally, SEI characterization was performed post-mortem with XPS, ssNMR and SEM to gain better insights into the underlying mechanisms. With 2 M LiTFSI and at -3.2 V, both the FE_{NH_3} (<22%) and R_{NH_3} (< $16 \text{ nmol s}^{-1} \text{cm}^{-2}$) remained relatively low. The SEI resembles a thin ($\sim 1\text{-}7 \mu\text{m}$) porous layer of Li ethoxide species, which is commonly known as a poor transport regulator for the reactant species. The FE_{NH_3} (50%) and R_{NH_3} ($350 \text{ nmol s}^{-1} \text{cm}^{-2}$) increased significantly at more negative potentials (-3.7 V) and SEI forms a thick and dense SEI layer ($\sim 800\text{-}1000 \mu\text{m}$) that is predominately enriched with LiF. This indicates the existence of a strong correlation between the FE_{NH_3} and the LiF concentration. Thicker and denser SEI morphologies are also beneficial for the FE_{NH_3} , while we also link them to current instabilities beyond -4 V. Therefore, it is more beneficial to operate at moderate applied potentials (-3.7 V) for long-term operation. At 1 M LiTFSI, the overall trend between the E_{we} and Li-NRR performance is relatively similar, but there is an overall reduction in the FE_{NH_3} and R_{NH_3} over the entire potential range. This is related to the lower availability of $\text{Li}^+\text{-TFSI}^-$ contact ion pairs, leading to a lower concentration of inorganic species in the SEI. Hence, both the E_{we} and the Li^+ solvation environment play a key role in the eventual morphology and composition of the SEI. These findings improve the current understanding of the SEI formation process and showcases new optimization strategies for Li-NRR systems that contribute to the development of a sustainable ammonia production process.

2. It is not clear the explanation of the unstable behavior of the current at the lower potential (higher in modulus) in correlation with the observed analysis. The correlation of the SEI layer with salt concentration, and the one of the WE potential with the salt reduction, was reported.

We thank the reviewer for pointing out the missing explanation for the current instability at very negative potentials as was also pointed out by reviewer 1. We refer the reviewer to our response to comment 1 of reviewer 1 on page 1-2.

3. The preparation of samples for SS-NMR is not clearly explained.

We apologize for the unclear write-up. The method section is corrected in the new version, accordingly.

To further address the response of comment 3 we added the following to the Method section:

For ssNMR sample preparation, the entire SEI layer (7-17 mg) that was built-up during the potentiostatic measurements was scraped off from the WE with a PTFE spatula and placed inside a mortar. Between 37-47 mg of KBr was added to the mortar as an inert filler and was mixed with the SEI material with a pestle. The SEI-KBr mixture was carefully packed into a 3.2 mm diameter airtight ZrO₂ rotor. This rotor filling procedure was entirely executed in an Ar glovebox to prevent moisture and air exposure. All items that were used during the rotor filling procedure (including KBr) were pre-dried at 60 °C in a vacuum oven for at least 24 hours before they were introduced into the glovebox for the sample preparation.

4. Testing experiments lasts 4 h, which is a huge duration: the authors must demonstrate that NH₃ does not oxidize in this timeframe.

It was earlier demonstrated by Chorkendorff and coworkers that the electrolyte acidifies during the oxidation of THF due to the release of protons [*Electrochem. Commun.* 2022, 134, 107186], which means that ammonia will be present as ammonium cation (pK_a = 9.25). This also explains why we could barely detect any ammonium in the downstream acid trap during preliminary measurements. Oxidation of the ammonium cation has proven to be very difficult and might be related to poor adsorption in comparison with ammonia because it lacks a free electron pair [*Chem. Rev.*, 2009, Vol. 109, No. 6; *J. Electrochem. Soc.* 2007, 154, B263]. Therefore, we expect that only a negligible amount of ammonium might be lost due to oxidation. Additionally, long term experiments in a similar single-compartment autoclave cell executed by the MacFarlane and Simonov group have shown stable ammonia production rates and Faradaic efficiencies running up to 96 hours without showing any loss of ammonia product by oxidation [*Nature*, 2022, 609, 722-727].

5. Looking at Fig. 10, it does not seem that this work is well placed in the scientific community state of art. Why it should be published in a Nature journal?

The nature of this work is more fundamental and tries to investigate if any correlations exist between the applied potential, the current efficiency and ammonia production rate with the focus on SEI composition and structural analysis. We did not intend to obtain the highest performance metrics for Li-NRR, but the average current density at -3.7 V (184.2±6.0 mA cm⁻²) and the energy efficiency at -4.6 V (7.6±0.3 %) at a reasonable FE_{NH₃} and R_{NH₃} over the course of 4 hour measurement time are actually among the best Li-NRR systems in the literature (see Supplementary Fig. 25 and Supplementary Table 1).

We are convinced that our results and obtained insights have a high degree of novelty and are relevant for the ammonia community and to the broader scientific audience of *Nature Communications*.

To be more aligned with the storyline, Fig. 10 and Supplementary Fig. 25 were interchanged from the manuscript to the Supplementary Information and *vice versa*.

To further address the response of comment 5 we interchanged Fig. 10 with Supplementary Fig. 25 in section "Energy efficiency of a batch Li-NRR system".

Supplementary Fig. 25. The literature entries from the batch systems (indicated in blue) include an additional energy input for sacrificial THF oxidation. The blue ellipsoidal area represents the spread of our results ranging from -3.1 V to -4.6 V using 2 M LiTFSI in 0.1 M EtOH/THF as electrolyte. For the batch cell literature entries, we used ref ^{2,3,4} at 1 bar, ref ⁵ at 10 bar, ref ⁶ at 15 bar, ref ^{7,8,9,10} at 20 bar and ref ^{11,12,13,14} for the continuous flow cell. All data is summarized in Supplementary Table 2.

6. When testing the 3 potential values, the authors study the Li⁺ permeability: it is fine, but the most critical part is the N₂ permeability (which is a more relevant limiting factor). Could you address this point?

We thank the reviewer for pointing out the missing N₂ permeability in our main analysis. Literature discussing the N₂ permeability is added into the introduction, and we used it to improve our interpretation and analysis of the results.

To further address the response of comment 6 we added the following paragraph to the Introduction section (unchanged text is indicated in black):

Theoretical work suggests that the Li-NRR elementary reaction steps are fast due to the very negative potentials applied for Li plating below -3 V versus the standard hydrogen electrode (SHE), meaning that the diffusion of reactant species (Li⁺, N₂ and H⁺) through the bulk and the SEI are the rate limiting step.^{12, 20} When studying the reaction at atmospheric N₂ pressure, the ammonia faradaic efficiencies (FE_{NH₃}) and production rates (R_{NH₃}) are limited by the N₂ transport due to its low solubility.²¹ This issue is typically circumvented by operating at higher pressures in an autoclave cell or by implementing a gas diffusion electrode to minimize the diffusion boundary layer thickness.^{4, 12, 22, 23} The N₂ flux must however be balanced by the H⁺ flux (related to the EtOH concentration) to prevent excessive lithium nitridation (H⁺ limited regime) or EtOH hydrolysis (N₂ limited regime). Additionally, the diffusion rates of both N₂ and H⁺ should not be significantly lower than Li⁺ to prevent unselective lithium deposition. Hence, to reach the optimal conditions for the Li-NRR, one must carefully balance the transport rates of the reactant species by optimizing the reactant concentration and the properties of the SEI.

To further address the response of comment 6 we added the following paragraphs to section "Potential effect on the composition and morphology of the SEI":

Based on the earlier mentioned equilibrium potentials, we expected that all species in the electrolyte would decompose at -3.2 V and form a mixture of both organic and inorganic SEI species, while the XPS results clearly indicate an EtOH-derived layer. We propose that the EtOH decomposition reaction on Li^0 is preferred because it is the most kinetically unstable compound in the electrolyte. Consequently, the thin Li-alkoxy passivation layer formed upon electrode polarisation can hinder the decomposition reaction of other species in the electrolyte.³⁶ Although ammonia production was observed, the low FE_{NH_3} (< 20%) indicates that nitrogen activation is not favourable at these conditions, which is most likely related to an imbalance in the transport rates of Li^+ , H^+ , and N_2 diffusion through the SEI layer. Since our operating conditions (N_2 pressure, EtOH and salt concentration) are relatively similar to previous literature reports observing higher R_{NH_3} and FE_{NH_3} 's, we expect that the properties of the SEI at -3.2 V is mainly responsible for the low FE_{NH_3} . Additionally, Spry *et al.* and Benedek *et al.* operated at ambient N_2 pressures and managed to obtain similar FE_{NH_3} 's (~20%),⁴⁰ and even higher (40%).⁴¹ Hence, increasing the N_2 operating pressure does not directly result in better Li-NRR performance. In partial agreement with the work of Chorkendorff, Norskov and coworkers,²⁰ it seems that the SEI properties have a much stronger effect on regulating the transport rate of Li^+ and H^+ than on N_2 . The porous Li-ethoxide structure seems to be particularly poor in slowing down the Li^+ and H^+ diffusion rate, leading to build-up of more SEI material (due to rapid and uncontrolled Li plating) and the formation of hydrogen gas. The latter explains the visible macro pores and cavities in the internal segment (Supplementary Fig. 12b) and top surface of the SEI (Supplementary Fig. 12c).

We find a clear correlation between the FE_{NH_3} and the LiF concentration in the SEI (induced by the potential driving force). LiF is not necessary to make ammonia, but simulations based on first principles have pointed out that the Li^+ transport resistance within LiF layers is much higher in comparison to other SEI species.²⁰ Therefore, the Li^+ diffusion rate (and perhaps also the H^+ diffusion rate as was pointed out by us earlier) slows down relative to N_2 transport and this increases the lithium nitridation probability, leading ultimately to higher FE_{NH_3} . These findings match well with our experimental observations, but the structure of our SEI is more complicated and most likely resembles a mixture of polyhetero organic and inorganic microphases that is typically observed in Li-ion batteries.¹⁷ Hence, we cannot exclude the influence of other species than LiF entirely.

Thicker and especially denser SEI structures obtained at -3.7 V and -4.6 V (in comparison with -3.2 V) seem to also correlate with the FE_{NH_3} . A similar observation was earlier reported by McShane *et al.*, wherein the FE_{NH_3} monotonically increases with the SEI thickness. We expect that the more tortuous paths from the bulk towards the electrode surface slow down the Li^+ (and probably H^+) transport rate, while N_2 is less affected. The exact mechanism at play remains unknown and requires further investigation. The current response is a direct measure of the Li plating rate and the Li^+ flux because Li plating is considered to be the only electrochemical reaction at the working electrode. Therefore, instabilities in the current response (Fig. 4a) could be related to a dynamic process between SEI thickening and breakdown, leading eventually into an increase or decrease of the Li^+ transport resistance during chronoamperometry measurements.

7. A long paragraph is present on LiF, however it is not corroborated by experimental evidence.

The last three paragraphs of the section "Potential effect on the composition and morphology of the SEI" form indeed a lengthy discussion on LiF. We agree with the reviewer that some parts of the previous write-up may be a bit biased towards correlating the LiF concentration to many different aspects of the Li-NRR performance metrics. The revised version of the manuscript has a more subtle wording and introduces also a new paragraph which discusses another possible correlation between the SEI morphology and the FE_{NH_3} .

Nonetheless, both XPS and ssNMR data show a clear increasing trend in the LiF concentration *versus* the overpotential, which can be interpreted as an indirect link with the FE_{NH_3} (because it increased with the overpotential). The same correlation was also observed by other research groups [*Nature*, 2022, 609, 722-727; *Joule*, 2022, 6, 1–19].

To further address the response of comment 7 we refer the reviewer to the revised version of section “Potential effect on the composition and morphology of the SEI” we provided as a response to comment 2 of reviewer 2 on page 13-15.